# Oxidative Stress Markers in Inflammatory Bowel Diseases: Systematic Review

**DOI:** 10.3390/diagnostics10080601

**Published:** 2020-08-17

**Authors:** Małgorzata Krzystek-Korpacka, Radosław Kempiński, Mariusz A. Bromke, Katarzyna Neubauer

**Affiliations:** 1Department of Medical Biochemistry, Wroclaw Medical University, Chałubińskiego 10, 50-368 Wroclaw, Poland; mariusz.bromke@umed.wroc.pl; 2Department of Gastroenterology and Hepatology, Wroclaw Medical University, Borowska 213, 50-556 Wroclaw, Poland; radoslaw.kempinski@umed.wroc.pl

**Keywords:** Crohn’s disease, ulcerative colitis, mucosal healing, antioxidants, lipid peroxidation, biomarkers

## Abstract

Precise diagnostic biomarker in inflammatory bowel diseases (IBD) is still missing. We conducted a comprehensive overview of oxidative stress markers (OSMs) as potential diagnostic, differential, progression, and prognostic markers in IBD. A Pubmed, Web of Knowledge, and Scopus search of original articles on OSMs in IBD, published between January 2000 and April 2020, was conducted. Out of 874 articles, 79 eligible studies were identified and used to prepare the interpretative synthesis. Antioxidants followed by lipid peroxidation markers were the most popular and markers of oxidative DNA damage the least popular. There was a disparity in the number of retrieved papers evaluating biomarkers in the adult and pediatric population (*n* = 6). Of the reviewed OSMs, a promising performance has been reported for serum total antioxidant status as a mucosal healing marker, mucosal 8-OHdG as a progression marker, and for multi-analyte panels of lipid peroxidation products assessed non-invasively in breath as diagnostic and differential markers in the pediatric population. Bilirubin, in turn, was the only validated marker. There is a desperate need for non-invasive biomarkers in IBD which, however, will not be met in the near future by oxidative stress markers as they are promising but mostly at the early research phase of discovery.

## 1. Introduction

Inflammatory bowel diseases (IBD) are chronic, idiopathic, and complex diseases of the gastrointestinal tract. Their two most common forms are ulcerative colitis (UC) and Crohn’s disease (CD). IBD pathogenesis is not fully elucidated but is believed to encompass genetic, immune, and environmental factors which together lead to the disruption of a delicate homeostasis between host immunity and the digestive tract microbiome [1].

Despite being the focus of attention in the field of translational medicine in the last two decades, precise diagnostic biomarkers for IBD are still missing. The time from the first symptoms to the final diagnosis of IBD has not shortened during recent years and even the exponential progress made in the understanding of the IBD pathogenesis has not changed the position of endoscopy, which, however invasive, remains the main tool in patient diagnosis and management [2].

Moreover, the incidence and prevalence of IBD in recent years is on a rise. For illustration, it is expected that, in Canada, the prevalence rate will increase from 0.5% in 2008, through 0.7% in 2018, up to 1.0% in 2030 [3]. The epidemiology of IBD is best established in developed countries: in North America and Europe, over one million and two million people are affected by IBD, respectively [4]. The prevalence of IBD exceeded 0.3% in North America, Oceania, and many countries in Europe and is the highest in Europe (UC: 505 per 100,000 in Norway; CD: 322 per 100,000 in Germany) [5]. The incidence of UC and CD per 100,000 person-years in North America and North Europe in estimated over 7.71 and 6.38, respectively. Most studies report a stable or decreasing incidence of IBD in North America and Europe and a rising incidence in newly industrialized countries in Africa, Asia, and South America [5]. Additionally, we are witnessing a change in the age profile of IBD patients [6]. Epidemiological data state clearly that there is a growing incidence of IBD among the elderly population. In Sweden, more than 20% of newly diagnosed patients are older than 60 [7]. Amon the elderly, the time to diagnosis is even longer than in younger adults, and any delay in the proper diagnosis may negatively affect the disease outcome. Elderly patients, usually suffering from multiple comorbidities, could particularly benefit from non-invasive testing [8]. Non-invasive diagnostic tools are of even more importance for pediatric IBD patients, and yet, the limited discovery of biomarker in pediatrics prompted Shores and Everett [9] to claim children to be “biomarker orphans”.

Oxidative stress, defined as an imbalance between prooxidants and antioxidants, is tightly associated with inflammatory responses and, as such, it has been implicated in the propagation and exacerbation of IBD. The infiltration of mucosal tissue with activated phagocytic immune cells generating reactive oxygen and nitrogen species (ROS and RNS, respectively) causes a shift towards prooxidants. It disturbs cellular homeostasis by damaging key macromolecules and contributes to cell injury and increased permeability of mucosal barrier, thus accelerating and perpetuating the ongoing inflammation. Recently, it has been suggested that oxidative stress may also be implicated in IBD pathogenesis as several oxidative stress-relevant genetic risk loci associated with IBD have been identified. It is indisputably a main trigger of neoplastic transformation in IBD patients [10].

The ROS and RNS, usually highly reactive and unstable molecules, are generated through the inflammation-mediated up-regulation of various lipoxygenases (LOX), myeloperoxidase (MPO), and inducible isoforms of nitric oxide (NO) synthase (NOS2), cyclooxygenase (COX2), and NADPH oxidase (NOX2). For the cellular homeostasis to be preserved, the activity of prooxidants has to be counterbalanced by antioxidants. The first line of defense encompasses antioxidant enzymes such as superoxide dismutase (SOD), catalase (CAT), glutathione peroxidase (GPx), peroxiredoxins, and paraoxonase (PON1), the main role of which is preventing the formation of free radicals and the neutralization of those already formed. Further, it includes transitional metal-chelating proteins such as transferrin, ceruloplasmin, and albumin which are tasked with sequestering free iron and copper and preventing them from participating in Fenton reaction. The second line consists of free radical scavengers, which neutralize free radicals by donating electrons, including glutathione (GSH), uric acid, cysteine, bilirubin, carotenoids, and vitamins A, E, and C [11,12,13]. The third and fourth lines of defense are focused on the removal of the damage done by prooxidants at the molecular and cellular level, respectively, and will not be addressed in this review. All main macromolecules are susceptible to oxidative damage which may have many forms and yield various primary and secondary reaction products. As they are more stable, and, in case of secondary products, they tend to accumulate, they are preferentially evaluated as the surrogate markers of oxidative stress.

Our goal was to provide an overview of oxidative stress markers, understood as prooxidants, antioxidants, and markers of oxidative damage to macromolecules, emerging during last two decades as potential biomarkers in IBD to address the question whether they are ready to support diagnosis and management of IBD patients.

## 2. Materials and Methods

To review all studies measuring oxidative stress markers in IBD patients we have searched three publication databases: PubMed, World of Knowledge and Scopus. Combinations of following keywords: (“oxidative stress” or “oxidant stress”) AND (“Crohn’s disease” or “ulcerative colitis” or “inflammatory bowel disease *” or “IBD”) AND (“* marker *” or “index” or “indices”) were used in queries. The asterisks allowed us to retrieve records where query words appeared with prefixes and suffixes (e.g., bio|marker|s). The search was limited to publications published between 1 January 2000 and 26 April 2020. No language restrictions were applied, although reports and publications in languages other than English were filtered out in following curation steps. Duplicate records from the databases were removed prior to first eligibility screening. Exclusion criteria were as follows: experimental studies (including animal studies and in vitro research), non-IBD, non-original articles, not on biomarkers; not on oxidative stress, and non-English language. The same criteria were applied for abstract and full-text screening, at which steps, however, also studies on genetic polymorphisms and interventional studies without control, non-IBD group, and between-group comparison of baseline parameters were removed. Two authors (MAB and RK) conducted all literature searches. Two authors (MKK and KN) separately reviewed the abstracts and based on the selection criteria, decided the suitability of the articles for inclusion. All authors then reviewed the eligible articles. References of the selected papers were cross-searched for omitted relevant articles. Three authors were contacted, of which two responded.

The following data were retrieved from reviewed publications: name of investigated biomarker and methodology; source (serum, plasma, saliva, urine, tissue, breath); IBD phenotype (CD or UC or mixed cohort); study population (adult/pediatric); type of control group (healthy individuals/patients with irritable bowel syndrome (IBS)/non-IBD patients with various gastrointestinal symptoms (GIS); number of patients in general and those with active disease; applied IBD activity scoring system and corresponding cut-off for defining active disease; main findings of the study such as level in analyzed groups and correlation with clinical (disease activity, location, extension, behavior, and complications) and biochemical data (erythrocyte sedimentation rate (ESR); C-reactive protein (CRP), interleukin (IL)-6, fecal calprotectin); markers characteristics if available (area under receiver operating characteristics (ROC) curves (AUC) and/or sensitivities and specificities with corresponding cut-off value, and/or positive and negative predictive values (PPV and NPV) and/or positive and negative likelihood ratios (LR+/LR−), correlation coefficients. To allow for a better comparison between studies reporting markers sensitivities and specificities, we calculated Youden index, a single measure combining both parameters, using the following equation:(sensitivity [%] + specificity [%]) − 100. (1)

Analysis of data was conducted according to PRISMA recommendations.

## 3. Results

Our primary search using the selected key phrases resulted in 874 publications, of which 507 entered the title screening phase. Exclusion criteria removed 298 entries leaving 209 records in our database for abstract screening. After excluding 110 papers, the full texts of 99 articles were read. Cross-search and manual search revealed additional 12 eligible articles. Authors of three papers have been contacted to provide full text of their article (*n* = 2) or details on their findings, of which two responded. As such, in one case, the reported data are retrieved from abstract. Finally, this systematic review was prepared based on 79 publications (Figure 1).

Most of eligible articles did not report data allowing for the evaluation of potential marker performance such as AUC, sensitivity and specificity or odds ratios. Collected data on analyzed compounds for which some marker characteristics was available, depending on markers possible application, are presented in the tables in sections: Section 3.1, Section 3.3 and Section 3.4 (based on 15 articles). All found markers, including those having non-quantified potential are presented separately (Appendix A). In Appendix A, they have been grouped by their classification as pro-oxidants/stressors (Appendix A), antioxidants, with an additional stratification into enzymatic antioxidants (Appendix A), non-enzymatic protein antioxidants (Appendix A), low-molecular weight antioxidants (Appendix A), vitamins and related compounds (Appendix A), or as markers of oxidative damage to macromolecules, subdivided into lipid peroxidation markers (Appendix A), markers of oxidative damage to proteins (Appendix A), and finally, markers of oxidative damage to DNA (Appendix A).

The majority of the markers have been evaluated in both IBD phenotypes and they were not specific for either UC or CD. Therefore, their performance is discussed in UC and CD at the same time to keep the comprehensive character of the review.

### 3.1. Interpretative Synthesis of Data: Diagnostic Markers

The vast majority of studies was designed to compare the level of oxidative stress markers between healthy individuals and IBD patients and thus to assess their potential suitability as diagnostic markers. Currently, the diagnosis of IBD is based on a combination of clinical presentation, endoscopic tests, and histologic examination and, specifically in CD, other imaging modalities such as computed tomography and magnetic resonance. Single diagnostic marker is missing and at present, the available laboratory tests play only a supplementary role [14]. Among multiple serological, immunological, genetic, and microbiological indices evaluated in IBD, only the performance of C-reactive protein (CRP) and fecal calprotectin (FC) was good enough to use them in clinical practice. However, a lack of standardized cut-off value of FC and unsatisfactory specificity remain its major limitations [15]. Unsurprisingly, the ability to distinguish diseased patients from healthy individuals was the most frequently examined trait in reviewed papers. Unfortunately, it was rarely explored further with an actual evaluation of the power of association and its diagnostic utility.

#### 3.1.1. Diagnostic Markers in IBD (Markers Not Specific for Either UC or CD)

From among prooxidants (if not otherwise stated, reviewed in Appendix A), the most frequently evaluated was nitric oxide, a protoplast of all RNS, to which four papers have been dedicated. However, three of them come from the same research group and were published within the three years and it is very likely that the populations examined there are overlapping [16,17,18]. Nonetheless, NO has been shown to be elevated in saliva [16,17,18] and mucosal tissue [19] of IBD patients, both in CD and UC, and hold promise as a general IBD marker.

Various measures of general prooxidant capacity were also frequently assessed in serum/plasma but did not yield consistent results [20,21,22,23], either due to differences in methodologies or populations or, more likely, both. As discussed by Boehm et al. [21], peroxidative potential largely depends on the amount of polyunsaturated lipids, which is decreased, particularly in active CD, and may cause earlier depletion of peroxidation substrates than in individuals with a better nutritional status, thus resulting in paradoxically lower peroxidative potential in patients. Also Cu, as an indicator of IBD presence did not yield consistent results, with one study showing its elevation in IBD patients as compared to controls [24], one demonstrating significant association exclusively among females [25], and the last one not finding an association [26]. Prooxidant enzymes such as MPO (E.C. 1.11.2.2.) in serum [27] and mucosal spermine oxidase (but exclusively in inflammatory cells; E.C. 1.5.3.16) [28], COX2 (E.C. 1.14.99.1) [29], NOX2 (E.C. 1.6.3.1) [29], and NOS2(E.C. 1.14.13.39) [29,30] have been unanimously elevated in IBD.

Three authors have calculated the so called “oxidative stress index” (OSI), obtained by dividing total oxidant capacity by total antioxidant status (TOC/TAS), and found it to be elevated in both CD and UC as compared to healthy controls in adult population [20,23] but not among children [22]. Yuksel et al. [20] found OSI to be an IBD predictor with odds ratio of 4.6 (Table 1).

Among enzymatic antioxidants (if not otherwise stated, summarized in Appendix A), superoxide dismutase (SOD; E.C. 1.15.1.1), catalase (CAT; E.C. 1.11.1.6), and glutathione peroxidase (GPx; E.C. 1.11.1.9), the triad of first line defense antioxidants, has been evaluated most often. There is a large discrepancy in the results regarding GPx and CAT activity in IBD. A drop in erythrocyte GPx has been indicative of IBD with 73% accuracy and that of CAT with 63% [31] (Table 1). However, others found erythrocyte CAT to be increased in UC [32] or unaltered in pediatric CD [33] and GPx to be unaltered in both adult IBD [34] and pediatric CD [33]. The activities of intracellular CAT and GPx determined in leukocytes obtained from an adult and a pediatric population, have been decreased and unaltered, respectively [35,36,37]. The activity of extracellular GPx isoform has been observed to drop [38] or increase [39,40] in IBD. In studies evaluating CD patients, GPx has been increased [39], increased only in active disease [41], or remained unaltered [42,43]. CAT activity has been also measured in tissue homogenates, but no significant differences between CD and UC patients and healthy controls have been observed [44], similarly to GPx determined in saliva [38]. Interestingly, some authors have measured plasma or serum activity of catalase although, unlike for SOD and GPx, the extracellular isoforms of the enzyme have yet to be reported [26,40].

Contrary to SOD, GPx, and CAT, available data on the activity of paraoxonase 1 (PON1) are consistent, at least regarding enzyme arylesterase activity. Boehm et al. [45] reported a drop in PON1 in both CD and UC patients, indicative of their presence with 74% and 65% accuracy, a finding corroborated by Yuksel et al. [20], who found PON1 to be an independent predictor of IBD (Table 1). Only recently these observations were supported by Sahin et al. [46] measuring arylesterase activity in UC patients (Appendix A). Collected data is less consistent regarding enzyme activity towards paraoxon [47] (Table 1) [46,48] (Appendix A), what should not come as a surprise, as none of the authors analyzed PON1 phenotype distribution. Unfortunately, analyzing PON1 paraoxonase activity without accounting for enzyme phenotypes is of limited value. PON1 is an enzyme with multiple functionalities, which, regarding its antioxidant activity, is responsible for protection against lipid peroxidation. A number of polymorphisms of PON1 have been described and one, Q192R, has a striking effect on enzyme activity towards paraoxon (paraoxonase activity; E.C. 3.1.8.1) but not towards phenyl acetate (arylesterase activity; E.C. 3.1.1.2). As such, arylesterase activity is believed to reflect enzyme concentration. In turn, the activity towards paraoxon ought to be assessed in groups stratified into an A phenotype of PON1 (homozygotes with Q at 192 and characterized by low paraoxonase activity) and AB/B phenotype (heterozygotes and homozygotes with R at 192, characterized by high enzyme activity) [49].

Among non-enzymatic protein antioxidants (if not otherwise stated, reviewed in Appendix A), the most popular one was albumin, although its concentrations have usually been reported in addition to other compounds and rarely information other than its levels in particular study groups has been provided. Except for two reports [47,50], serum/plasma albumin has been observed to drop in IBD, regardless the disease type and activity [20,51,52,53,54]. Also, albumin concentration in saliva has been significantly reduced [17]. Some authors have found it to drop only in active disease, either because patients in remission had its concentrations at the levels observed in healthy individuals [24] or because only patients in active phase had been enrolled [25,55]. Low albumin concentration is one of the predictors of malnutrition [54] and its decrease in IBD patients, particularly in the active phase of the disease, is likely to reflect the disease-associated worsening of nutritional status. The similar function of an indicator of nutritional status can be fulfilled by transferrin, which also decreases solely in IBD patients with active disease [56] (Table 1). Indeed, the active disease is among independent factors associated with increased risk of malnutrition in IBD patients [57]. Malnutrition in IBD is common and has a complex, multifactorial background, including altered nutritional intake, malabsorption, medication, excessive gastrointestinal loss, and changed energy requirements. It is considered as one of the extra-intestinal manifestations of IBD and affects particularly pediatric and elderly patients [58]. Albumin has only one redox-active thiol group from cysteine (Cys-34), and due to its high concentration it accounts for 80% (500 µmol/L) thiols in plasma [59]. Nimse et al. [12] described albumin as a “sacrificial protein” as its role as an antioxidant is associated with preventing ROS and RNS from attacking essential plasma proteins on the expense of the sole free thiol group of albumin. The oxidation of the human serum albumin decreases its half-life and increases its clearance by hepatocytes [60]. The loss of albumin due to oxidative damage can be used to explain negative relation of albumin to the IBD severity observed in several studies (see Appendix A). Upon the action of ROS, a protein cysteine can be monooxidated to sulfenic acids (Cys-SOH). Further oxidation may lead to the formation of sulfinic acid (Cys-SO_2_H) and sulfonic acid (Cys-SO_3_H). The latter two represent irreversible oxidation states and are often associated with pathological oxidative stress [61]. Albumin in which Cys-34 thiol group has been oxidized is called non-mercaptalbumin, whereas reduced Cys-34 (-SH) is a hallmark of mercaptalbumin. As determined by Nakatani et al. [62], 20–25% of human serum albumin is in form of non-mercaptalbumin-1, that is albumin in which Cys-34 is oxidized by being reversibly bound to various small thiol compounds. In non-mercaptalbumin 2, the Cys-34 is irreversibly oxidized to sulfinic or sulfonic acid [60]. The main free thiol of human plasma is Cys [63], what is in agreement with the fact, that Cys-S-S-Protein (a mixed disulfide) is the most abundant form of protein-bound thiols in human plasma [64]. On the basis of kinetic data reported by Bocedi et al. [65], the pool of HSA-Cys-SOH (formed in reactions with, e.g., H_2_O_2_) can be reduced to a mixed disulfide HSA-Cys-S-S-Cys by free cysteine. The mixed disulfide can be further reduced to mercaptalbumin (HSA-Cys-SH) by another molecule of cysteine. It is worth to mention, that GSH which is the main small-molecule thiol in erythrocytes (concentration of 1 mM) is present in plasma at levels more than 20 times lower than cysteine (~4 µM) [63] and thus, seems not to play a role in the maintenance of protein-thiol redox potential. Under pathologic conditions, the level of oxidized albumin may increase up to 70% [66,67]. Moreover, oxidation of albumin has an impact on its binding and transport capabilities. There are two sites located at the surface of albumin (multi metal binding site or site A), which may bind transition metal ions, such as Cu, Ni, and Co. For instance, HSA carries about 15% of human blood copper [68]. The oxidized HSA has reduced capacity to bind these metal ions, which in turn may easily catalyze chemical reactions generating free radicals [68]. Copper in whole blood and serum was observed to be positively correlated with the CD activity indices (Appendix A).

From among low-molecular antioxidants (if not otherwise stated, reviewed in Appendix A), the most commonly assessed was, so called, total antioxidant status or total antioxidant capacity (respectively, TAS or TAC), a cumulative measure reflecting a sample’s power to resist the prooxidant action, to which all present antioxidants are believed to contribute. If measured in serum or plasma, it seems to have been almost unanimously decreased in IBD, both CD and UC and regardless very diverse methodology, but exclusively in adult patients [20,23,37,40,69,70,71] (Table 1 and Appendix A). Koutroubakis et al. [70] evaluated TAS as well as a parameter referred to as corrected TAS, in which the contribution of albumin, uric acid, and bilirubin has been subtracted. Only Pereira et al. [71] observed elevation of TAS levels. Sampietro et al. [72] reported that surgical intervention restored TAS to the levels observed in healthy individuals. It is worth mentioning, that TAS has only recently been found to be a good marker of mucosal healing, as will be further discussed in detail in a section of the review dedicated to the mucosal healing [73] (Table 1). Interestingly, those promising results do not translate into pediatric IBD, which, also consistently, have not seemed to affect TAS concentrations at all [22,35].

The main contributors to TAS, apart from albumin, are bilirubin and uric acid. When assessed individually, total bilirubin has been consistently and in accord with the results on TAS, reported to be decreased in IBD [10,53,74,75,76] (Table 1 and Appendix A) with an exception of Şen et al. [50], who found bilirubin in CD patients to be elevated. As bilirubin is a standardized biochemical parameter, it has omitted several early steps in the process of marker development. Even though bilirubin has been evaluated in the IBD context for the first time only recently, there are already several papers providing the quantified assessment of its association with IBD (Table 1).

Fewer studies have been dedicated to uric acid, another important TAS component and a standardized biochemical parameter. However, unlike for TAS and bilirubin, their results are contradictory. Serum uric acid (SUA) has been determined in IBD patients in four and its saliva level in one paper. It has been found to be decreased in three studies [17,53,73] but to be increased in the remaining two [77,78]. Tian et al. [77] calculated that for the highest SUA quartile, the risk for UC manifestation is increased 1.2-fold (Table 1 and Appendix A). Although the association is rather weak, it was further confirmed by a large study of Zhu et al. [78], who analyzed SUA adjusted to creatinine (SUA/Cr) to account for the potential interference from the renal system. The authors have also observed an elevation of SUA/Cr in patients either with CD or UC, as compared to controls. Even the role played by uric acid is ambivalent. On one hand, it is one of the most important serum/plasma antioxidants, claimed to scavenge more than a half of free radicals in the circulation, while on the other hand, its elevation disturbs metabolic balance and is a risk factor for cardiometabolic diseases.

Some of enzymatic antioxidants rely on macro- and micronutrients such as Zn, Fe, Cu, Se and Mn for their activity. Of those, however, Cu and Fe, if not protein-bound, interact with hydrogen peroxide and produce hydroxyl radical in Fenton reaction, exacerbating rather than attenuating the oxidative stress. Accordingly, an increased Cu/Zn ratio accompanies inflammatory and oxidative stress conditions [79]. Although the evidence on the role of micronutrients in the pathogenesis of IBD is limited, two large prospective studies have demonstrated that zinc intake is associated with decreased risk of CD but not UC [80]. Malnutrition accompanying some IBD patients, particularly those with CD, may either cause or exacerbate the existing deficiencies. Indeed, the estimates on the prevalence of zinc deficiency among IBD patients differ from 14% to even 40% [79]. Our literature screening retrieved only a few papers, in which Se and Zn have been assessed. Moreover, they have usually been measured as an addition to enzymatic antioxidants. It stands to reason that there are more articles, which, however, would be more likely to be retrieved by search terms associated with malnutrition and nutritional indices than oxidative stress markers. Nonetheless, two studies have failed to demonstrate zinc association with the disease presence [24,26] and in one [25] its concentrations have been reduced in IBD patients. Additionally, one study has evaluated prognostic value of zinc, which will be discussed in details in the appropriate section [81]. Four studies have analyzed selenium and in only one of them have micronutrient concentrations been decreased in IBD patients as compared to healthy individuals [82], while others have failed to demonstrate significant differences [24,43,83].

Vitamins like vitamin A (retinol), E (α-tocopherol), and C (ascorbic acid) play a role of second line defense antioxidants, which act by scavenging free radicals and thus are responsible for preventing initiation and propagation of peroxidation processes. Health benefits are also thought to be provided by carotenoids, particularly β-carotene, lycopene, lutein, and zeaxanthin, and mainly associated with their high capacity for electron donation. In addition, β-carotene as well as α-carotene and β-cryptoxanthin serve as precursors for the synthesis of vitamin A [84,85]. Vitamin C is chemically capable of reacting with most of the physiologically important ROS and RNS and acts as a water-soluble antioxidant. Together with vitamin E, they are believed to play a crucial role in protecting lipid membranes from peroxidation. Similarly, as in the case of micronutrients, malnourished IBD patients may suffer from vitamin deficits, the phenomenon which might be used for diagnostic purposes.

Among vitamins and related compounds (if not otherwise stated, reviewed in Appendix A), carotenoids have been studied most often. Except for one study on each reviewed carotenoid [43,86,87], published data seem to indicate decrease in α- [24,43] and β-carotene [24,41,43,87,88], lutein, and zeaxanthin [24,87] among IBD patients. Also, the summary measure of all carotenoids [24,87], as well as the concentrations of lycopene and β-cryptoxanthine [24,43,87], have been consistently demonstrated to be lower in IBD. Retinol concentrations have been shown to be lower in both CD and UC [72,86,87], which, however have not been confirmed by all authors [43,86]. Data concerning vitamin C have shown it to be decreased in CD or IBD in general, either significantly [24,43] or insignificantly [88,89]. The most controversy concerns vitamin E, with two studies showing its IBD-associated drop [72,87], one demonstrating an insignificant decrease [88], and in the remaining four, no association at all [24,43,86,89].

Lipid peroxidation is a process in which double carbon-carbon bonds are attacked by ROS, particularly hydroxyl and hydroperoxyl radicals, resulting in formation of lipid peroxide radicals (LOO•), hydroperoxides (LOOH•), and conjugated diens, referred to as primary products of lipid peroxidation. After the process is initiated, the propagation phase is ensued and, unless disrupted by antioxidants, cannot be terminated until the final lipid peroxidation products are formed. The sensitivity of lipids to undergo peroxidation is directly proportional to the number of double bonds; therefore, polyunsaturated fatty acids (PUFAs) are the most susceptible. Uncontrolled peroxidation of lipids leads to cellular stress and damage of cells, tissues and organs. Among the most popular secondary products of lipid peroxidation, formed through spontaneous degradation of lipid peroxides, are reactive aldehydes, malonodialdehyde (MDA), and 4-hydroxy-2-nonenal (HNE). MDA is believed to be more carcinogenic and HNE—more toxic [90]. Others secondary lipid peroxidation products include gaseous alkanes, such as etane and pentane, and isoprostanes, of which the most popular is 8-iso-PGF2α [91]. Lipid peroxidation markers are of particular interest in biomarker research as they have been successfully determined in urine and in exhaled air, which are more easily available for testing and in larger quantities than blood.

From among secondary lipid peroxidation markers (if not otherwise stated, reviewed in Appendix A), MDA has indeed been studied most often. Four reports of reduction [26,39,71,92] are contradicted by 17, in which the accumulation of MDA has been observed, either in tissue [44], erythrocytes [34], saliva [16,17,37], or serum/plasma [21,27,36,37,38,40,41,48,72,86,93,94].

Of the other secondary products of lipid peroxidation, our search retrieved also one paper assessing HNE, the authors of which reported its elevation in both CD and UC [93], and five evaluating 8-iso-PGF2α. Analyzed in urine, 8-iso-PGF2α has been found unaltered in pediatric [33] and significantly increased in adult CD patients [95]. Also in adults, serum/plasma concentrations of 8-iso-PGF2α have been unanimously found to be increased, in both CD and UC [43,96,97]. Gaseous alkanes, determined in exhaled air, have been elevated in CD as reported by Wendland et al. [43]. They were also core components of the diagnostic model devised by Monasta et al. [98] (Table 1) for differentiating IBD patients from non-IBD controls and characterized by very high accuracy, as discussed in the section dedicated to the differential markers. Circulating lipoproteins are susceptible to lipid peroxidation as well, especially in conditions associated with the diminished activity of PON1. Boehm et al. [21] measured the level of LDL oxidation by determining directly the concentration of oxidized LDL (oxLDL) and indirectly, by measuring the concentrations of antibodies directed against them (anti-oxidized LDL autoantibodies; oLAB). While oLAB concentrations were similar in patients and controls, oxLDL were unexpectedly lower in CD, what has been explained by the authors by concurrently reduced concentrations of cholesterol, related to poor nutritional status of CD patients. This finding as well as its cause has been corroborated by Grzybowska-Chlebowczyk et al. [22] who reported a drop in active IBD patients as compared to inactive ones as well as a positive correlation with total cholesterol concentration. Taken together, those observations imply that, unlike in cardiometabolic diseases, oxLDL is a poor candidate biomarker for conditions associated with malnutrition as it reflects nutritional status rather than oxidative stress.

Regarding primary products of lipid peroxidation, conjugated diens have been assessed only in two small studies. Serum concentrations of conjugated diens between patients and control have been found comparable, although following infliximab therapy they decreased [55], while their tissue accumulation was significantly accelerated in both CD and UC [44]. Lipid peroxides have been shown to be elevated in erythrocytes and plasma from, respectively, UC [17] and CD patients [41], but to be unaltered in plasma of children with CD [33]. In turn, lipid hydroperoxides have not differed significantly between CD patients and controls as reported by Boehm et al. [21], but were found significantly decreased in both CD and UC as observed by Dudzińska et al. [26]. The maximal rate of plasma oxidation, a parameter reflecting the amount of substrates available for peroxidation, has been comparable between CD patients and healthy controls [72]. Taken together, secondary rather than primary, lipid peroxidation products hold promise as future markers in IBD.

Oxidation of proteins may be less spectacular than lipid peroxidation but, due to their abundance, proteins are main intra- and extracellular targets for oxidative stressors. Oxidative modification depends on a stressor type and may concern any amino acid side-chain or the protein backbone, yielding a variety of products. Oxidatively modified proteins may experience loss, and less often gain, of function, are prone to unfolding, fragmentation, and cross-linking and aggregation, display altered interactions with their ligands and other proteins, have modified turnover frequently developing resistance to proteolysis and accumulating, and become targets for the immune cells [99].

From among markers of oxidative damage to proteins (if not otherwise stated, reviewed in Appendix A), the database search resulted in finding five reports on advanced oxidation protein products (AOPP). Krzystek-Korpacka et al. [51] and Baskol et al. [27] were first to report the accumulation of AOPP in CD and UC, although as a general marker of IBD, the performance of AOPP was unsatisfactory (Table 1). Their observations were subsequently confirmed by others in adult [69,92] and in pediatric IBD patients [33]. Knutson et al. [100] analyzed the levels of nitro- and chloro-tyrosine in mucosal biopsies as well as serum, and found their accumulation to be significant solely in serum. Chloro-tyrosine has been elevated in both CD and UC but nitro-tyrosine only in UC. However, earlier study of Keshavarzian et al. [19] showed nitro-tyrosine to significantly accumulate in bowel tissue from CD and UC patients. Protein carbonyls, formed by proteins with reactive aldehydes, were found by those authors to accumulate as well [19]. Unlike in mucosa, their concentrations in sera of IBD patients were unaltered [74]. Two authors have recently examined ischemia-modified albumin (IMA), ROS-induced modification to albumin occurring under hypoxic conditions, and found it to be elevated in IBD [52,101]. Kaplan et al. [52] calculated that an increase in IMA by one unit projects 1.5-fold increased risk of IBD manifestation (Table 1).

Although over 20 various DNA base lesions caused by oxidative insult have been known, the most widely studied is 8-oxo-2′-deoxyguanosine (8-OHdG), which is mutagenic and results in G→T substitutions. However, apart from mutations, oxidative modifications to DNA bases may cause replicative block, deletions, microsatellite instability, and changes at epigenetic level. Moreover, oxidized bases may interfere not only with replication but also with gene expression at transcriptional level. Regardless the route, the effect of oxidative DNA damage for cell welfare is deleterious [102].

Indeed, from among markers of oxidative damage to DNA (if not otherwise stated, summarized in Appendix A), 8-OHdG was evaluated most often and was the only marker within this group which was actually assessed with respect to its diagnostic value. Unsurprisingly, due to the close relationship between oxidative DNA damage and cancer, 8-OHdG usefulness for the surveillance of IBD patients has been tested (discussed in detail in an appropriate section). Nonetheless, 8-OHdG as well as other markers of oxidative DNA damage may also be potentially useful in IBD diagnosis and monitoring the effectiveness of treatment. D‘Odorico et al. [87] found 8-OHdG to be significantly elevated in leukocytes of both CD and UC patients and affected by treatment as its accumulation was substantially reduced in patients treated with steroids or immunosuppressants as opposed to treatment with 5-aminosalicylic acid alone. The enhanced accumulation of 8-OHdG in CD has been subsequently confirmed by Beltrán et al. [36]. The analysis of mucosal samples of IBD patients has revealed also accelerated accumulation of HNE-derived etheno-DNA adducts such as 1,N6-ethenodeoxyadenosine (εdA) and 3,N4-ethenodeoxycytidine (εdC) [103]. The literature search retrieved three accounts on single strand breaks in the DNA structure, the most common type of DNA damage in cells, the frequency of which have been found to be increased in UC [23] and CD in adult [88], but not pediatric patients [35].

#### 3.1.2. Diagnostic Markers in UC

Regarding prooxidant enzymes, arginase-1 mRNA has been elevated in UC patients in contrast to CD patients [29]. Similarly, intracellular SOD was up-regulated in UC [32].

Finally, two proteins with antioxidant functionality with differential expression in IBD patients and healthy individuals were reported: elevated serum metallothionein in UC [69] and mucosal prohibin 2, down-regulated in UC [104].

The authors analyzing performance of bilirubin, being the main contributor to TAS, in UC, reported the lowest quartile of total bilirubin to be associated with 2-fold and 6-fold higher risk of UC manifestation as demonstrated for the learning and validation cohorts, respectively. The results obtained for UC have been subsequently corroborated by another group, which reported 2.6 times higher likelihood of UC manifestation in individuals within the lowest bilirubin quartile [77].

Contrary to uric acid, described above, glutathione (GSH) role is definitely positive, as it acts as a protector against ROS and RNS, both directly, by scavenging them, and indirectly as a cofactor for a number of enzymes, GPx and others. Still, the reviewed reports are not consistent on the issue of its level in IBD patients in reference to healthy individuals. In UC, a GSH decrease in serum has been reported by Homouda et al. [69] and by Rana et al. in erythrocytes [32].

#### 3.1.3. Diagnostic Markers in CD

Among prooxidants, enzymesarginase-1 mRNA has been down-regulated in CD patients [29]. Further, intracellular SOD, determined in erythrocytes being enzymatic antioxidants, has been unaltered in adult IBD [31] but subsequent studies have demonstrated its counterintuitive up-regulation in CD [35]. Reports on pediatric CD have also been inconsistent with erythrocyte enzyme activity being either unaltered [35] or down-regulated [33]. The discrepancy might result from the fact that Koláček et al. [33] examined exclusively children in inactive phase of the disease while Pácal et al. [35] included probably also children with active disease, although the proportion of inactive-to-active was not specified. Similarly, the activity of extracellular SOD has been shown to be decreased [25] or unaltered [26]. Szczeklik et al. [42] observed enzyme down-regulation only in CD patients with active disease while Achitei et al. [38], on the contrary, observed this solely in patients with inactive IBD. Beltrán et al. [36] analyzed SOD activity in leukocyte homogenates and found it to be significantly up-regulated in CD patients in the active phase of the disease but normalized when those patients achieved remission. In the same study, enzyme activity in a separate group in inactive disease had was comparable to that of controls and significantly lower than in patients with active CD. Enzyme activity determined in saliva did not show any CD-related differences [42].

The last of the antioxidant enzymes retrieved using the search criteria was ceruloplasmin, assessed both as a protein [26] and as a ratio between its ferroxidase activity and apoceruloplasmin [55], of which only the latter has been found to drop in CD patients as compared to healthy controls.

It is worth mentioning, that from among low-molecular antioxidants the diagnostic power of TAS in CD has been demonstrated to be excellent and far better than that of C-reactive protein, one of the two biochemical indices used in clinical practice in the management of IBD patients [37]. Also TAS levels evaluated in saliva decreased in CD as reported by Jahanshahi et al. [16] and Rezaie et al. [17], what, however, was not corroborated by Szczeklik et al. [37].

Lenicek et al. [74] calculated that per 1 mmol/L reduction in total bilirubin, one of the main contributors to TAS like described earlier, a risk for CD manifestation increases by 13%. In turn, Schieffer et al. [76] evaluated that for the lowest quartile of total bilirubin the likelihood of CD manifestation was 1.9-fold higher in a learning set of patients and, even more elevated, 3.6-fold higher, in a validation cohort.

In CD, plasma/serum GSH has been found to be increased in pediatric population [86] or unaltered [88], decreased exclusively in active disease [37] (Table 1) or decreased exclusively in patients with complications in adult population [35]. One study on GSH in saliva from CD patients has shown its concentrations to be diminished but solely in active disease [37]. The levels of cysteine have been determined once, in CD patients, and have been found to be significantly decreased [88].

From among lipid peroxidation markers, Boehm et al. [21] found MDA to be a very good marker of CD, characterized by 91% overall accuracy in discriminating CD patients from controls, which, at optimal cut-off had very good specificity combined with satisfactory sensitivity and high positive likelihood ratio (Table 1). Similarly, good performance was noted for MDA as an active CD marker, with an overall accuracy of 87%, in a study of Szczeklik, 2018 [37] (Table 1). All of the above-mentioned studies have used the thiobarbituric acid (TBA) method for MDA determination. It has to be mentioned, that the method is unspecific as TBA reacts also with other aldehydes, sugars, biliverdin and bilirubin, referred together as TBA-reactive substances (TBARS). As such, MDA separation using high-performance liquid chromatography is highly recommended, as has been conducted by Akman et al. [34].

Selected oxidative stress markers with diagnostic potential are presented in Table 1.

### 3.2. Interpretative Synthesis of Data: Differential Markers

#### 3.2.1. Crohn’s Disease and Ulcerative Colitis

The differential diagnosis of two main IBD types may pose a problem when CD is restricted to the colon and at the time of diagnosis displays only inflammatory behavior without abscesses, fistulas or stenosis, thus closely resembling UC. Visible blood in feces is a key symptom of active UC, present in ca. 95% of patients, but it occurs in about half of CD patients presenting with a UC-like clinical phenotype of the disease as well. It is estimated that up to 14% patients are initially misclassified and their diagnoses have to be changed over time as the disease behavior alters into structuring or penetrating one, characteristic for CD. While CD location is relatively stable and rarely progress from exclusively colonic to ileocolonic, the disease behavior changes in up to on third of patients. The proper classification of patients is of paramount clinical relevance as it allows for employing a tailored clinical management, optimal treatment strategies, both in terms of pharmacotherapy and surgery, and for proper prognostication. The cost of improper diagnosis is a delay in introducing optimal therapy and repeated endoscopic examinations [105]. Therefore, it is not surprising that non-invasive biomarkers able to support the differentiation between CD and UC are sought.

Still, even though a number of reviewed papers reported differences in analyzed oxidative stress markers between CD and UC patients, the strength of association was tested with ROC analysis only in two. Boehm, et al. [45] analyzed arylesterase activity of paraoxonase (PON1) in plasma and found it to better differentiate CD and UC than CRP but the overall accuracy as well as sensitivity of the enzyme were poor despite good specificity (Table 1). Monasta et al. [98], in turn, analyzed VOCs in alveolar breath of children with IBD and found the presence of 13 to differ between CD and UC. The model based on those compounds, including reactive nitrogen species, reactive aldehydes, gaseous alkens, and child’s age was characterized by remarkably high (93%) accuracy and ability to correctly classify 87% of children (Table 1). This finding is of importance as it regards children, thus a group, which would particularly benefit from a non-invasive testing. Although very promising, the excellent performance of VOCs still needs to be confirmed and validated on a larger cohort.

A metaproteomic analysis of aspirates of mucosal-luminal interface conducted on pediatric IBD population by Zhang et al. [106] identified ROS and RNS-generating enzymes NOS2, LOX15 (E.C. 1.13.11.33), and dual oxidase 2 (E.C. 1.6.3.1)to be overexpressed in UC, regardless the bowel region from which the samples were harvested (Table 3 and Appendix A). Protein antioxidants such as hemopexin (Appendix A) and enzymatic antioxidants such as SOD, thioredoxin-dependent peroxidase reductase (E.C. 1.11.1.24), and ceruloplasmin (E.C. 1.16.3.1) (Appendix A) were increased in CD but solely in the mucosa obtained from descending colon and terminal ileum. The same antioxidants in ascending colon were up-regulated in UC. Also, upregulated in UC, regardless of the location of examined sample, was peroxirodoxin 2 (Appendix A). Still, the results of Zhang et al. [106] seem to indicate pronounced oxidative imbalance in mucosal tissue from descending colon and terminal ileum of children with UC, in whom the expression of oxidative stressors is elevated, and antioxidant defenses diminished. In the ascending colon, overexpression of prooxidants seems to be balanced by an up-regulated expression of antioxidants. However, the collection of aspirates remains an invasive technique, hence those interesting findings shedding a new light on oxidative stress in IBD do not translate into non-invasive biomarkers.

Additional oxidative stress markers, the levels of which differ between CD and UC could be found among low and high molecular weight antioxidants such as Se, demonstrated to be more diminished in CD than UC [83] (Appendix A) and albumin, found to be lower in CD than UC as well [20] (Appendix A). Those findings agree well with poorer nutritional status of CD patients. As already mentioned, the incidence of malnutrition in IBD is high [57] but is more prevalent among CD patients. It is associated with the disease location, as CD may affect ileum and interfere with nutrient absorption, and with differences in lipid and carbohydrate metabolism, characterized by higher rates of lipid oxidation and lower of carbohydrate oxidation as compared to UC patients and healthy individuals [107].

Also markers of oxidative damage to macromolecules such as 8-iso-PGF2a [96] (Appendix A), IMA [101] (Appendix A), and DNAssb [71] (Appendix A) occurred to be differently accrued. While 8-iso-PGF2a has been more markedly accumulated among CD patients, the concentrations of IMA and the level of oxidative DNA damage was significantly higher in UC. Also the number of arginase 1 transcripts in erythrocytes has been shown to be more markedly elevated in patients with active UC than active CD [29] (Appendix A). Although arginase-1 does not directly possesses prooxidant capacity, it competes with nitric oxide synthase for their substrate, arginine, and, if elevated and under inflammatory and oxidative stress conditions, causes the uncoupling of NOS enzymes, which switch to superoxide anion production [108]. As all those were analyzed in blood, either in serum or in peripheral blood mononuclear cell (PBMCS), they can be measured relatively non-invasively and therefore follow-up studies verifying their potential as differential markers would be welcomed.

#### 3.2.2. IBD and Other Gastrointestinal Disorders

The differential diagnosis of IBD is broad, and includes, but is not limited to, diseases like infectious enterocolitis, microscopic colitis, intestinal tuberculosis, celiac disease, colorectal cancer, non-steroidal anti-inflammatory drugs-associated enteropathy or irritable bowel syndrome [109]. Wide spectrum of diseases, from malignant and infectious to functional, which can manifest with similar symptoms like IBD, makes the diagnostic process complicated. It requires an armamentarium of investigation tools and procedures, and, in consequence, is time-consuming and expensive.

Of the reviewed papers, an attempt to compare the oxidative stress markers between patients with IBD and other diseases of the gastrointestinal tract that may potentially interfere with proper diagnosis has been undertaken in seven, of which only Monasta et al. [98] employed ROC analysis to determine diagnostic accuracy of examined markers. In their study, a panel of 15 VOCs and children’s age were capable of correctly classifying 94% of IBD patients with 65% specificity (Table 1). Again, it is a promising result but in need of confirmation and validation.

The already mentioned recent study of Zhang et al. [106], in addition to diagnostic and CD-UC differential markers, identified also proteins differently expressed between pediatric IBD and non-IBD. Interestingly, the authors examined not only human, but also microbial proteins and found microbial GPx to be overexpressed in aspirates sampled from IBD patients as compared to those harvested from non-IBD patients presenting with gastrointestinal symptoms justifying colonoscopy (Table 2). Among other identified proteins, but of human origin, were ceruloplasmin (Appendix A) and prooxidant NOS2 and LOX5 (E.C. 1.13.11.34) (Appendix A), all overexpressed in aspirates derived from IBD patients. In adult population, Keshavarzian et al. [19] observed elevated levels of NO (Table 2 and Appendix A) and accelerated protein carbonylation and formation of nitro-tyrosine (Table 2 and Appendix A) in inflamed and non-inflamed mucosa from UC patients with clinically active disease as compared to individuals presenting with abdominal pain and/or positive tests for fecal occult blood but with normal mucosa on endoscopic examination. Ozhegov et al. [110], in turn, demonstrated that CD patients had elevated levels of mucosal indices of free radical processes and increased rates of peroxide radical generation and susceptibility to peroxidation than patients with irritable bowel syndrome, a functional bowel disorder (Table 2 and Appendix A). They were also having, probably as a compensatory mechanism, an increased total antioxidant capacity (Table 2 and Appendix A).

Unlike in mucosal tissue, the level of TAS in plasma of IBD patients (Appendix A) has been significantly decreased as compared to patients with non-inflammatory and non-neoplastic gastrointestinal disorders [111]. Similarly decreased have been the levels of total bilirubin, GSH, and albumin (Table 2) [111]. However, Grzybowska- Chlebowczyk et al. [22] analyzed TAS (Appendix A) as well as TOC and OSI (Appendix A) and lipid peroxidation markers oxLDL and oLAB (Appendix A) and compared them between children with IBD and without but presenting with gastrointestinal symptoms and did not observe significant differences. In only one paper, the levels of investigated analytes have been compared between patients with IBD and those with adenomas and adenocarcinomas. Starczak et al. [89] analyzed vitamins A, E and C in plasma (Table 2 and Appendix A) and 8-OHdG in leukocytes (Table 2 and Appendix A) and demonstrated that IBD and CRC patients differ with respect to their vitamin A and C and 8-OHdG concentrations, all significantly higher in IBD [89]. As all those indices may be assessed in a non-invasive manner, a follow-up with an appraisal of their diagnostic value as differential markers in IBD against, respectively, functional bowel disorders and CRC is warranted.

Selected oxidative stress markers with differential potential are presented in Table 2.

### 3.3. Interpretative Synthesis of Data: Markers of Disease Progression (Activity, Severity, Mucosal Healing, and Colorectal Cancer)

Assessment of the disease activity is crucial for the choice of optimal therapy. Despite the fact that several scoring systems are available, they are far from being perfect. Scales incorporating the clinical symptoms and selected laboratory indices, such as Crohn’s disease activity index (CDAI), do not reflect the intestinal manifestation as there is a rather poor correlation between the symptoms and endoscopic findings [112]. Moreover, the multiplicity and diversity of available scales, as well as inconsistencies in their naming, make it difficult to compare the research results, even more so as their authors frequently forgot to state the applied scoring system. The situation is further aggravated by the lack of uniform cut-off values as well as by authors’ neglect in their reporting. As such, patients with the same severity of the condition may be differently classified. The confusion is additionally intensified by the frequent lack of data on the proportion of patients in remission or in the active phase of the disease in the studied populations, the fact of crucial importance for the findings to be of any value. What also came as a surprise was that not all authors analyzed the possible correlation between evaluated markers of oxidative stress and the disease activity, even though there was evidence that such data had been available.

All those factors listed above are likely to contribute to the observed discrepancies between studies and are significant “roadblocks” in biomarker discovery and translation of basic research into clinical practice. There are also differences in the treatment regimens between evaluated cohorts and one has to keep in mind that most of drugs used in the management of IBD patients are likely to directly or indirectly affect the reductive-oxidative balance and consequently the oxidative stress markers. Still, not always those regimes have been revealed in the reviewed articles and rarely their possible effect examined. Therefore, this part of the review was based only on the articles reporting data concerning the disease activity, its extent, and possible effect of treatment.

#### 3.3.1. Progression Markers in IBD (Markers Not Specific for Either UC or CD)

Keshavarzian et al. [19] reported mucosal NO to be higher in CD and UC patients in clinically active than inactive phase of disease and to increase linearly along the following sequence: non-IBD controls with gastrointestinal tract symptoms, inactive IBD, IBD of mild activity, IBD of moderate activity, IBD of severe activity.

Arylesterase activity of PON1 was more markedly down-regulated in active CD and UC than inactive diseases, inversely and moderately correlated with their respective scores of clinical activities, that is, CDAI and Mayo disease activity index (MDAI), and with markers of inflammation severity [45] (Table 1). Erythrocyte GPx activity measured by Krzystek-Korpacka et al. [31] was markedly down-regulated in patients with active CD or UC and was inversely related to CDAI and MDAI as well as to inflammatory markers, and had similar power to CRP in discerning patients with active IBD (Table 3).

Selected oxidative stress markers with potential to assess progression of IBD are presented in Table 3.

Additional discussed markers or indices can be found in supplementary tables. From among non-enzymatic protein antioxidants (Appendix A), thiols have been reportedly more markedly depleted in patients with active than inactive IBD [34] and UC, in which they were inversely correlated with clinical activity of the disease expressed in terms of Truelove-Witts score [58]. The association between thiols and IBD activity has been confirmed by Neubauer et al. [73], who also reported negative correlation between their concentrations and CDAI/RI and inflammatory indices as well as lack of difference between active and inactive phases of diseases (Table 3). Transferrin, similarly to albumin a negative phase reactant and an index of nutritional status, was also inversely related with the disease activity, being significantly decreased exclusively in CD and UC patients in active phase of the disease. Transferrin levels displayed negative correlation with coefficients for CDAI and RI, respectively, and other inflammatory indices [56] (Table 3).

No significant association with the disease activity have been reported for Zn and Se, except for one study [82] in which Se concentrations decreased along with UC severity and corresponded with its extension (Appendix A). Serum/plasma TAC has been inversely related to inflammatory indices in CD [33,37,72,73] and UC [23,73], although those observations have not been confirmed by all authors [40,71]. It has been also shown by some to be more markedly diminished in active CD [37,71,111] and inversely correlated with indices of its clinical activity [33,35,37,73,111] as well as inversely related to the disease extension in UC [70] (Table 1 and Appendix A).

In large studies evaluating total bilirubin as an IBD marker, its concentration has not been associated with the disease activity or extension [74,76,77] (Table 1). As discussed in Section 3.1, based on limited number of articles there is no consensus on uric acid in IBD. Nonetheless, those reporting its reduction in IBD have also shown its inverse relationship with clinical disease activity [53,73] (Table 3 and Appendix A).

No correlations with the disease activity or inflammatory markers have been reported for vitamin A, E, or C although vitamins A and E have been markedly more depleted in patients with BMI<20 [87] (if not otherwise stated, discussed results are in Appendix A). D’Odorico et al. [87] found β-carotene and lutein in CD, β-cryptoxanthine in UC, and zeaxanthin, lycopene, and total carotenoids in both CD and UC to be markedly diminished in plasma of patients with active than inactive disease, however, without correlation with clinical activity indices or inflammatory markers. Hengstermann et al. [24] corroborated findings on lycopene and observed borderline lower concentrations of luthein/zeaxantine in patients in active phase of IBD.

From among lipid peroxidation markers the concentration of oxLDL in pediatric IBD was, counterintuitively, lower in active phase of the disease [22]. As discussed earlier, is does not necessarily indicate the alleviation of oxidative stress but rather the depletion of substrates as malnutrition in IBD is tightly associated with the disease activity. Thus, with increasing IBD activity, oxidative stress and nutritional status of patients are operating in the opposite directions and the net effect is a drop in oxLDL observed by Grzybowska-Chlebowczyk et al. [22] as well as by Boehm et al. [21].

Regarding markers of oxidative damage to proteins, Keshavarzian et al. [19] reported accelerated protein carbonylation and formation of nitro-tyrosine in active CD and UC as compared to inactive phase of diseases as well as high positive correlation coefficients for their levels and the sequence: non-IBD controls with gastrointestinal tract symptoms – inactive IBD – IBD of mild activity – IBD of moderate activity – IBD of severe activity (if not otherwise stated, discussed results are in Appendix A). Enhanced nitro-tyrosine accumulation in sera of patients with active UC was in turn reported by Knutson et al. [100] who additionally examined chloro-tyrosine and found its formation to be associated with the disease activity as it was more markedly accelerated among patients with active CD and UC. Also enhanced accumulation of AOPP in serum/plasma has been reported in active phase of UC [92] or its positive correlation with CDAI and inflammatory indices [51] (Table 1), although others have failed to observe it [27]. Weak positive correlations with the disease activity indices CDAI and RI as well as with inflammatory markers have been reported also for ischemia-modified albumin [52] (Table 1).

#### 3.3.2. Progression Markers in UC

The content of NO in saliva has not correlated with the clinical activity of UC or the extent of the disease and was not affected by treatment [18] while that determined in mucosal tissue have shown high positive correlation (all summarized in Appendix A). The NO generation seemed to be tightly associated with clinical rather than endoscopic activity as it was elevated in patients with active UC as compared to inactive regardless whether tissue in the former was sampled from inflamed or non-inflamed areas of the colon [19]. Also the immunoreactivity of ROS-generating spermine oxidase in immune cells infiltrating the diseased colon has positively correlated with the clinical activity of UC as well as with endoscopic and histopathologic one [28]. In turn, the numbers of COX2, NOX2, and NOS2 transcripts in mucosa were significantly higher in patients with active than inactive CD or UC and that of arginase 1 – in active UC [29]. Serum MPO did not correlate with clinical activity or the markers of inflammation severity [27]. The TOC has been shown to have no correlation with clinical activity of UC but to positively correlate with severity of inflammation [23] or to positively correlate with clinical activity of both UC and CD [20].Blood concentrations of Cu have been elevated more pronouncedly in active IBD [24] and those measured in plasma have strongly and positively correlated with CRP [26].

In agreement with tight association between PON1 and both clinical activity and severity of inflammation, Sahin et al. [46] found arylesterase activity of the enzyme in UC patients to be predicted by MDAI score and leukocyte count (if not otherwise stated, discussed results are in Appendix A).

Transferrin has been evaluated as a marker of active UC and has been characterized by good accuracy with superior sensitivity to specificity (Table 1). Zhao et al. [75] evaluated bilirubin in UC patients and found its lower levels reflecting the disease clinical activity as well as severity of inflammation and the disease extension. From among lipid peroxidation markers serum 8-iso-PGF2a has been significantly higher in active than inactive UC [96] and higher in active than inactive IBD when measured in urine [97].

Regarding markers of oxidative damage to DNA none of the reviewed articles reported correlation between the disease activity, severity of inflammatory response or the disease extension except for positive correlation between DNA single strand breaks (DNAssb) and CRP and ESR in UC patients reported by Aslan et al. [23] (Appendix A) and 8-OHdG association with UC extension reported by D’Incà et al. [114] (Table 3).

#### 3.3.3. Progression Markers in CD

Plasma peroxidation potential measured by Boehm et al. [21] was unexpectedly lower in active than inactive CD and inversely correlated with CDAI and inflammatory indices, which, however, was probably caused by reduced availability and, thus, earlier depletion of peroxidation substrates associated with nutritional deficits in patients with active CD. Yuksel et al. [20] not only found that OSI positively and strongly correlated with CDAI in CD and with Rachmilewitz index (RI) in UC, but also demonstrated that this oxidative stress index was an independent predictor of both and displayed a positive although weaker correlation with inflammation severity in both CD and UC (Table 1).

Koláček et al. [33] observed a negative correlation between erythrocyte GPx and clinical activity of the disease in pediatric CD patients (pediatric CD activity index; PCDAI) and Pácal et al. [35] between PCDAI and enzyme activity in leukocytes. Concerning serum/plasma activity ofGPx, no associations were reported except for a positive correlation with fecal calprotectin, currently the most appreciated biochemical index in IBD clinics, noted by Vaghari-Tabari et al. [40] and, also counterintuitive, more marked reduction in the enzyme activity in patients with inactive than active IBD demonstrated by Achitei et al. [38]. However, one has to regard the results obtained for serum/plasma activities of GPx, SOD or catalase with caution. Those enzymes are mainly intracellular, the activity of which, even if detectable in serum/plasma, is comparatively lower than that within the cells, even by orders of magnitude, and thus their determination there is less reliable. Accordingly, the situation is repeated with catalase: its activity has been more pronouncedly down-regulated in active than inactive CD in the erythrocytes [31] (Table 1) and leukocytes [37] but to positively correlate with fecal calprotectin when measured in serum [40]. The erythrocyte activity of SOD has been inversely, although weakly, associated with CDAI and erythrocyte sedimentation rate (ESR) [31] in CD but not in UC [31,32] and correlated, also inversely, with fecal calprotectin in children with CD [33]. Plasma/serum activity of SOD has been, in turn, found either more markedly decreased among patients with active CD and inversely related to clinical disease activity and severity of inflammation [42] or, on the contrary, significantly more decreased in inactive than active IBD [38] and strongly positively correlated with CRP [26].

Bourgonje et al. [54] evaluated albumin-adjusted thiols exclusively in CD patients in remission and observed that their depletion is related to the disease extent, as the index was significantly lower in patients with ileocolonic involvement than colonic. Additionally, it was inversely related with the severity of inflammation. In general, albumin has also been reported to be lower in active IBD [24] or specifically in active CD [51,111]. Albumin inversely correlated with CDAI and CRP [46,111]. Somehow confusing results were reported by [25] showing decreased albumin in IBD patients, and in the same time its positive correlation with clinical disease activities.

In spite of in large studies evaluating total bilirubin as an IBD marker, its concentration has not been associated with the disease activity or extension [74,76,77] in others, however, more substantial drop has been seen among patients with active CD [111], with parallel increasing severity of the condition assessed using clinical activity scales [53,75,111] or expressed by severity of inflammation [53,75,111]. The bilirubin reduction extent matched patients’ classification into inactive CD – CD of mild activity – CD of moderate activity – CD of severe activity categories [53]. Contrary to all reports discussed above, Şen et al. [50] recorded a drop in total bilirubin in their CD patients with inactive disease and an increase in active form of the disease.

The index of SUA adjusted to creatinine, elevated in IBD, was in turn associated with CD activity by being both more markedly increased in individuals in active condition and by weakly and positively correlating with CDAI [78]. Zhu et al. [78] also observed SUA/Cr association with the disease location (Table 1). The last of evaluated low molecular weight antioxidants, GSH, has been more substantially diminished in active CD and has inversely correlated with CDAI and CRP, in both plasma [111] and in saliva [37]. The performance of GSH as an active CD marker has even been quantified and it has been characterized by good accuracy, superior over that of CRP [37] (Table 3). However, no such correlation could be found in pediatric population, in which there an opposite tendency has been reported: GSH tended to be higher in active than inactive CD [86].

In one of the two studies in which MDA was assessed as a diagnostic marker in CD, it has been accumulated markedly in active disease, positively correlated with CDAI, and had an excellent accuracy as an active CD marker, superior over CRP [37] (Table 3). However, according to the other study no correlation with CDAI could be found [21] (Table 1). Szczeklik et al. [37] observed MDA to correlate with CDAI and CRP also when the marker was measured in saliva (Table 3; if not otherwise stated, discussed results are tabularised in Appendix A). Of the remaining study reports evaluating MDA in IBD, no correlation with CD or UC has been reported and a few publications presented the marker to be inversely related to CRP and ESR [26,72]. From among lipid peroxidation markers other than MDA, only oLAB has shown a weak positive correlation with CDAI [21].

#### 3.3.4. Markers of Mucosal Healing

Therapeutic strategy in UC has evolved into a “treat to target” approach. The most commonly accepted therapeutic targets became endoscopic remission (defined as Mayo Endoscopic Score; MES≤1) and clinical remission, that is, patient-reported outcome (defined as resolution of rectal bleeding and diarrhea/altered bowel habit) [115,116]. Both current and novel treatment options are now evaluated based on their efficiency in the mucosal healing (MH) induction [117]. Significance of the MH results directly from its enormous impact on the outcome of this life-long disease. MH is associated with reduced risk of disease relapse, hospitalization, colectomy, and colitis-associated neoplasia [118]. An invasive endoscopic examination, repeated every 3-6-12 months, is considered a golden standard for MH diagnosis. Therefore, similarly to IBD diagnosis, biomarkers which could substitute endoscopy in its evaluation are intensively searched for. However, like in case of assessing clinical activity, various scales and different cut-offs are in operation and this lack of standardization causes difficulties in the interpretation and comparison of the results of clinical trials as well as basic and clinical research.

Among the reviewed articles, the association between investigated analytes and MH or inflammation was determined, by one way or the other, in 13 articles. Several groups [19,83,119,120] have compared compound levels between inflamed and non-inflamed mucosa, which has a scientific value in elucidating molecular mechanisms of the disease; however, being a tissue-based assessment, does not translate into a desirable biomarker. Nitro-tyrosine and protein carbonyls, both markers of oxidative damage of proteins, have been demonstrated to be significantly elevated in the inflamed mucosa [19] (Appendix A). Data concerning enzymatic antioxidants seem to be conflicting. The GPx activity (Appendix A), as well as the concentration of its cofactor glutathione (Appendix A), has been reported to drop in inflamed mucosa [83] while the number of enzyme transcripts to increase (Appendix A) [119]. The protein expression of peroxirodoxins either increased or decreased, depending on the family member. Further level of thioredoxin-dependent peroxide reductase decreased while the expression of SOD increased [120] (Appendix A). Increasing levels of prooxidant spermine oxidase in colonic mucosa in parallel to the increasing endoscopic and histopathologic activity of the disease has been observed as well [28] (Appendix A). In turn, Küçük et al. [104] reported weak positive correlation between endoscopic and histopathologic activity of UC and the immunoreactivity of antioxidant prohibin 2 (Appendix A); however, provided scatterplots do not support the notion. Other authors have investigated blood-based markers and reported their significant correlation with the disease endoscopic activity. Negative correlations have been noted for antioxidants such as catalase [31], transferrin [56], uric acid, free thiols, and TAS [73] (Table 3). Positive correlation was observed for total oxidative capacity of serum/plasma [20] (Appendix A) and for oxidative stress index (OSI) [20] as well as for markers of oxidative stress-related protein alterations, namely, IMA [52] (Table 1) and AOPP [92] (Appendix A). Additionally, IMA [52] and OSI [20] have been shown to be independent predictors of Rachmilewitz endoscopic activity index (Table 1). There was only one paper, in which oxidative stress markers have been analyzed as potential MH markers [73] (Table 3). SUA, F-SH, and TAS have shown moderate to good diagnostic power. The best performance characterized TAS, the accuracy of which was 87% and its specificity was excellent, the accompanying sensitivity, however, was not satisfactory.

#### 3.3.5. Markers of Colorectal Cancer

IBD patients have increased risk for colorectal cancer [121,122]. Therefore, according to the European Crohn´s and Colitis Organization guidelines, patients with UC, with the exception for individuals with proctitis, bearing a lower cancer risk, should regularly undergo surveillance colonoscopy aimed at detecting dysplasia and early cancer [121]. As dysplastic and neoplastic changes in UC may be flat, not well defined, or/and multifocal, their diagnosis requires multiple biopsies. It is recommended that during surveillance colonoscopy, per every 10 cm of the large bowel, four random biopsies are to be taken, yielding several dozen samples per patient [122]. Despite the intensive development of endoscopic techniques, like chromo-endoscopy, which allow taking targeted biopsies, it remains a challenge for both endoscopist and pathologist [123]. Therefore, cancer surveillance in IBD is yet another area where non-invasive biomarkers are urgently needed. Yet, only five authors have examined oxidative stress markers between UC patients with and without co-occurring neoplasms in a form of low- or high-grade dysplasia or UC-associated cancer (UCAC) (referred to as progressors and non-progressors, respectively). Moreover, only two of them [69,124] examined blood-based, and thus non-invasive, markers, while remaining three analyzed marker expression in the tissue [30,113,114]. It may help to understand the molecular mechanisms of UC-associated carcinogenesis but does not solve the problem of invasiveness of sample collection in potential application as markers of progression. Of the investigated oxidative stress-associated molecules, 8-Oxo-2’-deoxyguanosine, an oxidized derivative of deoxy-guanosine and the major product of oxidative damage to DNA, was evaluated in three studies and found to be more markedly accumulated in colonic mucosa among progressors than non-progressors in two. D’Incà et al. [114] observed a gradual increase in mucosal 8-OHdG along a sequence of normal-inflamed-dysplastic tissue. As a dysplasia marker, 8-OHdG was characterized by good (81%) overall accuracy and similar, moderate sensitivities and specificities and by high negative but poor positive predictive value (Table 1). If confirmed, monitoring 8-OHdG accumulation during surveillance colonoscopies might help to identify patients with higher risk of progression and allow for a less frequent endoscopic examinations in those ascribed to a low-risk group based on 8-OHdG. As D’Inca et al. [114] reported sublocation-related significant difference in 8-OHdG accumulation, Saigusa et al. [113] evaluated a large set of exclusively rectal tissues and confirmed, using immunohistochemistry, its suitability as a neoplasia marker in UC. At the cut-off selected by those authors, 8-OHdG has been characterized by high (89%) sensitivity but low specificity (Table 1). Still, Youden index calculated for both studies is similar (0.44 and 0.48). On the same set of tissues, Saigusa et al. [113] evaluated also 8-nitroguanine (8-NG), an indicator of nitrosative DNA damage, and found its performance to be superior to that of 8-OHdG. The 8-NG was characterized by both good sensitivity and specificity (83%; Table 1) and thus a higher Youden index (0.66). Nitrosative stress is perpetrated by reactive nitrogen species originating from nitric oxide (NO). Therefore, Saigusa et al. [113] investigated also the expression of inducible NO synthase (iNOS; NOS2) but found it to be only non-significantly up-regulated in progressors and therefore had a poor characteristics as a neoplasia marker in UC (Table 1). Corroborating observation regarding NOS2 but contradicting that on 8-OHdG, Kumagae et al. [30] found the markers to be equally high in both progressors and non-progressors (Appendix A, respectively). Indices reflecting oxidative damage to proteins, that is, protein carbonyls in colonic mucosa [124] and serum accumulation of advanced oxidation protein products [69] have been examined as well and found to be increased in progressors (Appendix A) but the strength of observed association has not been tested. Importantly, Chen et al. [124] observed that protein carbonylation is accelerated also in non-dysplastic tissue of progressors. Hamouda et al. [69] reported gradually diminishing antioxidant glutathione in sera from progressors, non-progressors and healthy individuals (Appendix A). Importantly, AOPP and GSH have been determined in patients with mild to moderate dysplasia, therefore hold potential as early dysplasia markers [69]. The concentrations of metallothionein a thiol-rich metal-binding protein involved in protection against metal toxicity, oxidative stress included, were, in turn, increased and comparable between progressors and non-progressors (Appendix A).

### 3.4. Interpretative Synthesis of Data: Prognostic Markers

Predicting the disease course is crucial for the stratification of patients and choice of the optimal treatment option. Known prognostic factors, like disease extension and behavior or age at onset, have supplementary value, and objective prognostic indices are missing [81].

The automatic searching of databases using predefined criteria did not result in any publications dedicated to oxidative stress markers as prognostic factors. One such paper, regarding zinc, has been retrieved through cross-reference/manual search. Siva et al. [81] analyzed the association between low zinc concentrations and adverse disease-specific outcomes among IBD patients using prospectively collected data from IBD registry. They have found that patients with abnormally low zinc (<0.66 mcg/ml) were more likely to be hospitalized (1.4-fold in CD and 2.1-fold in UC increased probability), undergo surgery (2-fold in CD and 1.6-fold in UC), and have the disease complications such as malnutrition, anemia, hemorrhage, bowel obstruction or formations of fistulas, abscesses, or strictures (1.5-fold in CD and 2-fold in UC) (Table 4).

Three CD phenotypes can be distinguished, that is, inflammatory, structuring and penetrating. They have been included in the Montreal classification, which, in addition to phenotype, stratifies patients by the age at diagnosis and the disease location. All of those factors are of prognostic value but determining CD phenotype requires endoscopy of upper and lower digestive tract as well as imaging tests like CT-enterography or MR-enterography. Moreover, one of the limitations of the Montreal classification is the fact, that, despite rather stable disease location, its behavior changes [109,125]. Still, some authors have investigated the association of analyzed markers with Montreal classification and three of them have observed differences in analyzed compounds with respect to CD behavior. Pácal et al. [35] found CD patients with any form of complications (abscess, fistula, or stenosis) to have more markedly accumulated MDA/TBARS (Table 4 and Appendix A) and reduced GSH concentrations (Table 4 and Appendix A) in plasma and Zhao et al. [75] observed significant decrease in serum total bilirubin in CD patients with penetrating disease (CD-B3) (Table 4 and Appendix A). Both GSH and bilirubin significantly contribute to the total antioxidant state of the plasma/serum. Therefore, it is rather unexpected that Pereira et al. [71] found it to be elevated in penetrating as compared to inflammatory (B1) phenotype. Yet, those authors, contrary to most others, have found TAS to be increased in the disease (Appendix A).

### 3.5. Interpretative Synthesis of Data: Adult vs. Pediatric IBD

It is estimated that about one out of five IBD patients experience first symptoms as a child or teenager. Similarly to adult population, incidence rates in pediatric IBD are increasing. However, the diagnosis might be more difficult due to atypical symptoms and extraintestinal manifestations, which may precede the gastrointestinal indications of the disease even by years. Moreover, the prevalence of indeterminate colitis in children ranges between 5% and 30% and is thus 2-fold higher than in adult patients. Indeterminate colitis in children is considered a temporary diagnosis, which makes it difficult to implement optimal therapy. Early onset of the disease is associated with a more aggressive behavior and complicated course. Repeated endoscopic examinations are particularly poorly tolerated by young patients. Therefore, it is a group, which would benefit from non-invasive testing above all others [126].

However, the development of new pediatric biomarkers is insufficient at best. Their discovery is even more complicated than that for adults, if only because of the difficulties with obtaining reference material, since the research on healthy children had to be limited to minimal risk procedures. This and many other reasons for this situation have been thoroughly discussed in the excellent reviews of Savages et al. [9] and Shores et al. [127]. Disparity in number of research on adult and pediatric biomarkers was clearly visible here as well. Even assuming that our search criteria were too selective and not all potentially relevant studies were included, it would affect the exact number of retrieved articles but not the proportion between studies on adult and pediatric populations. We did not constrain our search and, yet, out of 79 eligible articles only six were analyzing oxidative stress markers in children, which makes it less than 8% of the pool. Our review also shows that simple extrapolation of the biomarker results obtained in adults, without prior validation in pediatric population, is likely to be unsuccessful, as they do not follow the same trends. The general impression, based on evidence gathered here, is that adult IBD patients had oxidative stress “in full swing” with increased prooxidants, depleted antioxidants, and accelerated formation of products of oxidative damage to macromolecules. In children, the oxidative stress seems to be at an earlier stage of noticeable oxidative insult accompanied by the up-regulated antioxidant defenses. The study of Zhang et al. [106] provides an excellent illustration. The authors have conducted an analysis of both microbial and human mucosal proteome and showed that the up-regulation of both ROS- and RNS-generating enzymes such as NOS2, LOX5 and 15, and DUOX2 (Appendix A) is paralleled by an increase in antioxidant enzymes and proteins, that is, SOD, ceruloplasmin, and hemopexin (Appendix A). Moreover, they have also observed an up-regulation of antioxidants of microbial origin in response to the activity of host immune cells (Appendix A). The notion that research on adult biomarker might not translate well into pediatrics is, in turn, illustrated by TAS, TOC, and OSI parameters. TAS activity has been found diminished by most of authors and nearly all exceptions have come from pediatric research, where there have been no visible depletion of antioxidant capacity [22,33,35] (Appendix A). The TOC, TAS’s opposite, has been increased in adult studies [20,23], but unaltered in the only one pediatric study [22]. Consequently, the OSI index calculated on their basis, has been increased in adults [23] but unchanged in children [22] (Appendix A). Glutathione has been another antioxidant, which levels were elevated among pediatric IBD patients [86], while GSH measurement studies on adults have almost unanimously reported its depletion (Appendix A). Furthermore, the only study not confirming the depletion of β-carotene in IBD has been conducted on pediatric population [86] (Appendix A). Lipid peroxides have been reported to be elevated in IBD [32,41], but not among children [33]. Similarly, another lipid peroxidation marker, 8-iso-PGF2a, has remained not elevated exclusively in pediatric cohort [33] (Appendix A), just like a marker of oxidative damage to DNA [35] (Appendix A). In fact, only protein oxidation marker, AOPP, has been found to accumulate in children with IBD [33] like it did in adult patients (Appendix A).

As discussed above, both, the limited number of papers and the reported lack of association of markers seemingly promising in adults, clearly support the provocative thesis of Savage et al. [9] and Shores et al. [127] on children being “biomarker orphans”. However, the only two more sophisticated studies which were retrieved using the specified search terms, that is, metaproteomic analysis of aspirates of mucosal-luminal interface by Zhang et al. [106] and alveolar breath analysis by Monasta et al. [98], were conducted on pediatric populations. While the sample collection in the former one remains invasive, the breath analysis successfully implemented by Monasta et al. [98] is addressing the more restrictive requirements for pediatric biomarker for the “non-invasiveness” is a relative term regarding pediatric testing. The blood-based markers are considered non-invasive in adults but for an ill child a repeated blood sampling is less than unappealing. Therefore, other biological sources allowing for painless sample collection, such as saliva, urine or breath are preferable. It remains to be hoped that upcoming research will confirm and validate the diagnostic models devised by Monasta et al. [98] and we will see their implementation in the clinics of pediatric IBD soon.

## 4. Conclusions

Multidirectional research into the mechanisms of IBD has been rapidly growing in the recent years. Yet, this progress in knowledge is not evidenced in diagnostic algorithms, and endoscopy, invasive, expensive, and not necessarily accepted by patients, remains the main diagnostic tool in the hands of gastroenterologists. This review presents cumulative evidence supporting the notion on oxidative stress in IBD and its manifestation at both local and systemic level. It also shows that oxidative stress is expressed in all its forms, that is, by an accelerated oxidation, a depletion of antioxidant capacity, and by an enhanced formation of oxidatively modified macromolecules. It might not be entirely unanimous, and more evidenced regarding some markers than others, but the oxidative imbalance has been proven in the reviewed research. However, it has not yet been translated and is unlikely to be translated into accurate biomarkers that can be implemented in clinical practice soon. With this review, we attempted to summarize the current knowledge on oxidative stress-related biomarkers to define their position as diagnostic tools in IBD. We intended to use descriptive biomarker characteristics (accuracy, sensitivity, specificity, etc.) as one of the inclusion criteria. However, preliminary search showed that, disappointingly, it would leave us with only 17 articles. Even the studies summarized in Table 1 mostly reported results obtained on small cohorts and unconfirmed. Majority of the evaluated markers have been non- invasive and assessed in blood serum and cells, saliva as well as in exhaled air in contrast to indices studied in the intestinal tissues, what requires endoscopic biopsy. Some have shown promising marker characteristics, like TAS, GSH, and MDA as markers of CD in its active phase [37], 8-OHdG as a progression marker [114], and TAS, again, as a MH marker [73]. An excellent performance has been reported for the three diagnostic models developed based on differences in the composition of exhaled air [98]. Still, all those markers and models require validation. Bilirubin seems to be a welcome exception as, owing to its being a standardized biochemical marker of liver dysfunction, its association with IBD has been assessed by a number of studies including some large ones. Moreover, it has been quantified and the odds ratios calculated on learning cohorts have been subsequently validated on another large set of patients [76]. Oxidative stress by no means is a new concept and the research into it goes back further than the last two decades reviewed here. Thus, one might expect the research to progress from a discovery research to a validation phase. However, even with the most popular oxidative stress markers, we seem to be locked in an early stage of small studies conducted on poorly defined populations and offering knowledge of a marker being elevated/decreased in patients with active disease as compared to healthy individuals, with the resultant findings of limited clinical value. For a marker to be successful it has to improve the management of patients and there is a lot of room for improvement in IBD. Yet, only a few studies explored the potential of oxidative stress markers as prognostic, differential, progression, or MH/inflammation markers. In summary, oxidative stress markers in IBD have the potential to become the indices of clinical utility, however further studies conducted in the bigger cohorts and including detailed and descriptive marker performances are essential.

## Figures and Tables

**Figure 1 diagnostics-10-00601-f001:**
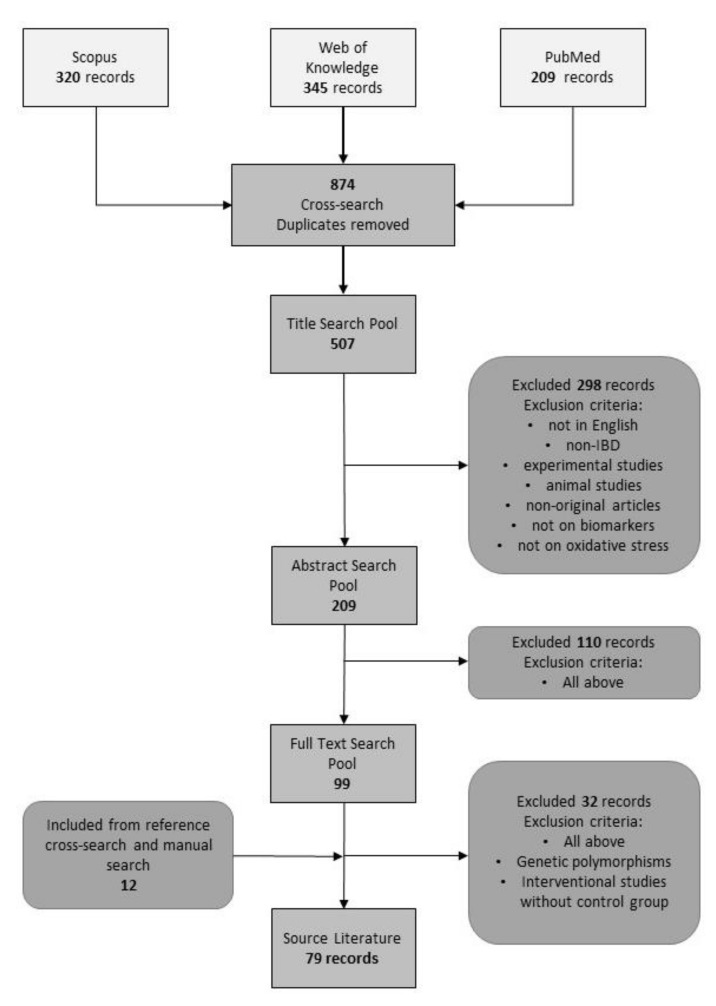
Flowchart presenting the selection process. IBD, inflammatory bowel diseases.

**Table 1 diagnostics-10-00601-t001:** Oxidative stress indices as potential diagnostic biomarkers in inflammatory bowel diseases.

Analyte	Type	Disease	Evaluated Population ^1^	Findings ^2^	Diagnostic Characteristics	Ref.
OSI	BS	IBD	40 CD (ns/CDAI, ns); 40 UC (ns/RI-EAI, ns); 80 HC	↑ in CD and UC vs. HC; CDAI 0.87; CRP ^3^ 0.39; RI-EAI 0.83; CRP ^4^ 0.40	IBD predictor: OR = 4.6 (2.7–7.7); predictor of log (CDAI) (B ± SE = 0.71 ± 0.06) and log (RI-EAI) (B ± SE = 0.6 ± 0.07)	[20]
PON1_A_ (E.C. 3.1.1.2)	BP	IBD	52 CD (37/CDAI ≥ 150) 67 UC (30/MDAI, ns); 99 HC	↓ in aCD and aUC vs. HC; ↓ in aCD vs. iCD; ↓ in aUC vs. iUC; CDAI −0.47; ESR^3^ −0.36; IL-6^3^ −0.45; MDAI −0.50; CRP^4^ −0.29; ESR^4^ −0.37; IL-6^4^: −0.45	CD marker: AUC = 0.74 ^6^; cut-off ≤ 122 U: sens. 48%, spec. 90% UC marker: AUC = 0.65 ^6^; cut-off ≤ 138 U: sens. 51%, spec. 75% Differential CD/UC marker: AUC = 0.63 ^7^; cut-off ≤ 101 U: sens. 35%, spec. 88%	[45]
PON1_A_ (E.C. 3.1.1.2)	BS	IBD	40 CD (ns/CDAI, ns) 40 UC (ns/RI-EAI, ns); 80 HC	↓ in CD and UC vs. HC; ESR ^4^: 0.40	IBD predictor: OR = 0.995 (0.992 0.997)	[20]
PON1_P_ (E.C. 3.1.8.1)	BP	CD	47 CD (28/CDAI ≥ 150); 23 HC	↓ in CD vs. HC; ↓ in aCD vs. iCD; CRP: −0.61	CD marker: AUC = 0.96; cut-off 378.25 U/L, sens. 94%, spec. 91%; Active CD marker: AUC = 0.87; cut-off 305.31 U/L, sens. 86%, spec. 84%	[47]
t-bil	BS	CD	90 CD (ns/ns, ns); 229 HC	↓ in CD; no association with Montreal classification	OR = 0.87 (0.82–0.92): each ↓ in t-bil by 1 mmol/L is associated with a 13% ↑ in the risk of CD manifestation	[74]
t-bil	BS	IBD	254 CD and 254 HC (learning) 187 UC and 187 HC (learning) 233 CD and 233 HC (validation) 124 UC and 124 controls (validation)	↓ in UC and CD vs. HC	For the lowest quartile of t-bil, an association with CD was OR: 1.91 (1.26–2.91) and with UC-OR: 1.98 (1.09–3.63) in the learning cohort and OR: 3.60 (2.19–5.99) and 6.07 (3.01–12.75) in the validation cohort	[76]
t-bil	BS	UC	170 UC (94/MDAI ≥ 3); 200 HC	↓ in UC; no association with UC activity	For the lowest quartile of t-bil, an association with UC was OR = 2.56, 95% CI: 1.54–4.25, *P* < 0.001	[77]
UA	BS	UC	170 UC (94/MDAI ≥ 3); 200 HC	↑ in UC; no association with UC activity	For the highest quartile of UA, an association with UC was OR = 1.20, 95% CI: 1.05–1.77, *P* = 0.045	[77]
MDA (TBARS)	BP	CD	52 CD (37/CDAI ≥ 150); 99 HC	↑ aCD and iCD vs. HC; IL-6 −0.31; no association with activity	General CD marker: AUC = 0.91 (0.85–0.95); cut-off 1.03 µM, sens. 75%, spec. 90%, LR+ 7.4 and LR− 0.28	[21]
aAOPP	BS	IBD	68 UC (33/MDAI, ns) 50 CD (38/CDAI, ns); 45 HC	↑in aUC and iUC vs. HC; ↑ in aCD vs. HC CDAI 0.42; ESR ^3^ 0.53; ESR ^4^ 0.35	IBD marker:AUC = 0.64 (0.56 − 0.72) ^3^; Cut-off: 1.67 µmol/g, sens. 52%, spec. = 80%, LR+ 2.6 and LR− 0.6	[51]
IMA	BS	IBD	35 UC (ns/RI-EAI ≥ 5) 33 CD (ns/CDAI ≥ 150); 65 HC	↑ in UC and CD vs. HC; ↑ in UC vs. CD CDAI 0.32; CRP ^3^ 0.30; ESR ^3^ 0.32 RI-EAI 0.31; CRP ^4^ 0.40; ESR ^4^ 0.43	1 unit of ↑ in IMA projects 1.5× ↑ risk of being IBD: OR: 1.5 (1.1–1.9) log(IMA) is an independent predictor of log (CDAI) (B ± SE = 2.33 ± 0.72) and log (RI-EAI) (1.82 ± 0.59)	[52]
VOCs	ΔB	IBD	34 pediatric CD (ns); 33 UC (ns); 65 GIS; 102 HC	Model for differentiating IBD from controls (GIS + HC): 18 VOCs + age: AUC 0.93 (0.89–0.96); could detect 96% of all IBD cases with 69% specificity Model for differentiating CD from UC: 13 VOCs + age: AUC 0.93 (0.88–0.99); 86.6% correctly classified cases, 94% sens. and 76% spec. in detecting CD Model for differentiating IBD from GIS: 15 VOCs + age: AUC 0.92 (0.87–0.96), identifies correctly 94% of IBDs, with a specificity of 65%		[98]

^1^ number of patients (number of patients with active disease/scoring system ≥ cut-off for active disease); ^2^ data presented as ↑ increased or ↓ decreased levels between indicated groups and as correlation coefficients preceded by the variable; ^3^ correlation in CD patients; ^4^ correlation for UC patients; ^5^ performance similar to that of C-reactive protein (CRP); ^6^ performance worse than that of CRP; ^7^ performance better than that of CRP; ns, not specified; NOS2, inducible nitric oxide synthase; OSI, oxidative stress index calculated as total oxidant capacity/total antioxidant status; PON1, paraoxonase-1 (A in superscript indicates the arylesterase activity and *P*—paraoxonase activity); t-bil, total bilirubin; MDA (TBARS), malondialdehyde determined as thiobarbituric acid-reactive substances; aAOPP, advanced oxidation protein products adjusted to albumin concentrations; IMA, ischemia-modified albumin; VOCs, volatile organic compounds; HC, healthy controls; CD, Crohn’s disease; UC, ulcerative colitis; IBD, inflammatory bowel disease; aCD, active CD; iCD, inactive CD; aUC, active UC; iUC, inactive UC; N, neoplasms; D, dysplasia; LGD, low grade dysplasia; HGD, high grade dysplasia; UCAC, UC-associated cancer, CRC, colorectal cancer; R, rectum; SC, sigmoid colon; GIS, non-IBD patients with gastrointestinal symptoms; CDAI, Crohn’s disease activity index, MES, Mayo endoscopic subscore; RI-EAI, Rachmilewitz endoscopic activity index; RI, Rachmilewitz index; HA, histologic activity; EA, endoscopic activity; FFPE, formalin-fixed paraffin-embedded; ESR, erythrocyte sedimentation rate; MH, mucosal healing; BES, Baron endoscopic score; T-I, tissue-based marker determined with immunohistochemistry (IHC); BS, blood-based marker determined in serum; BE, blood-based marker determined in erythrocytes; BP, blood-based marker determined in plasma; T-H, tissue-based marker determined in homogenates; ΔB, difference between alveolar breath and environmental air; AUC, area under receiver operating characteristics (ROC) curve; OR, odds ratios; B, regression coefficient; SE, standard error; sens., sensitivity; spec., specificity; PPV, positive predictive value; NPV, negative predictive value; LR+, positive likelihood ratios; LR−, negative likelihood ratios.

**Table 2 diagnostics-10-00601-t002:** Oxidative stress indices as potential differential biomarkers in inflammatory bowel diseases.

Analyte	Type	Disease	Evaluated Population ^1^	Findings ^2^	Application	Ref.
IoFRP	T-H	CD	45 CD (aa/ns); 30 IBS; IS	↑ in CD vs. IBS	Diff. CD/IBS	[110]
RoPRG	T-H	CD	45 CD (aa/ns); 30 IBS; IS	↑ in CD vs. IBS	Diff. CD/IBS	[110]
StP	T-H	CD	45 CD (aa/ns); 30 IBS; IS	↑ in CD vs. IBS	Diff. CD/IBS	[110]
LOX5 (E.C. 1.13.11.34)	T-M	IBD	25 pediatric CD (aa/PCDAI, ns); 22 pediatric UC (aa/PUCAI, ns); 24 GIS	↑ in IBD vs. GIS; (8.2× in AsC, 9× in DesC, 11.6× in TI) ^3^	Diff. IBD/GIS	[106]
NOS2 (E.C. 1.14.13.39)	T-M	IBD	25 pediatric CD (aa/PCDAI, ns); 22 pediatric UC (aa/PUCAI, ns); 24 GIS	↑ in IBD vs. GIS (16.3× in AsC, 9.5× in DesC, 5.3× in TI) ^3^ ↑ in UC in AsC (3.5×), DeC (3×), and TI (2.7×) ^3^	Diff. IBD/GIS Diff. CD/UC	[106]
NO	T-H	IBD	22 UC (15/ns), paired biopsies inf. & non-infl. colon (*n* = 6 aUC); 11 CD (6/ns); 14 specific colitis (infl.); 10 GIS	↑ in aUC and iUC vs. GIS; ↑ in aCD and iCD vs. GIS; ↑ in spec. colitis vs. GIS; ↑ in aUC/aCD vs. iUC/iCD ↑ in infl. and non-infl. aUC vs. iUC and GIS; If stratified by severity into: GIS/iIBD/miIBD/moIBD/sIBD, r = 0.81	Diff. IBD/GIS	[19]
CP_P_	T-M	IBD	25 pediatric CD (aa/PCDAI, ns); 22 pediatric UC (aa/PUCAI, ns); 24 GIS	↑ in IBD vs. GIS; (46× in AsC, 29× in DeC, 37× in TI) ^3^; ↑ in CD in DeC (1.4×) and TI (5.7×) but ↑ in UC in AsC (6.2×) ^3^	Diff. IBD/GISDiff. CD/UC	[106]
mGPx (E.C. 1.11.1.9)	T-M	IBD	25 pediatric CD (aa/PCDAI, ns); 22 pediatric UC (aa/PUCAI, ns); 24 GIS	↑ in IBD vs. GIS; 4× (AsC), 14× (DeC), 11× (TI) ^3^	Diff. IBD/GIS	[106]
Alb	BP	CD	55 CD (35/CDAI ≥ 150); 25 GIS	↓ in aCD vs. GIS; ↓ in aCD vs. iCD; CDAI −0.76; CRP −0.41	Diff. CD/GIS	[111]
TAS (FRAP)	BP	CD	55 CD (35/CDAI ≥ 150); 25 GIS	↓ in aCD vs. GIS; ↓ in aCD vs. iCD; CDAI −0.57; CRP −0.46	Diff. CD/GIS	[111]
TAS (S_IND_2)	T-H	CD	45 CD (aa/ns); 30 IBS; IS	↑ in CD vs. IBS	Diff. CD/IBS	[110]
t-bil	BP	CD	55 CD (35/CDAI ≥ 150); 25 GIS	↓ in aCD vs. GIS; ↓ in aCD vs. iCD; CDAI −0.52; CRP −0.48	Diff. CD/GIS	[111]
GSH	BP	CD	55 CD (35/CDAI ≥ 150); 25 GIS	↓ in aCD vs. GIS; ↓ in aCD vs. iCD; CDAI −0.76; CRP −0.41	Diff. CD/GIS	[111]
LOOH	T-H	CD	45 CD (aa/ns); 30 IBS; IS	↑ in CD vs. IBS	Diff. CD/IBS	[110]
Vit.C	BP	IBD	51 IBD (ns); 67 AD (90% tubulare); 136 CRC; 79 HC	↑ in IBD vs. CRC	Diff. IBD/CRC	[89]
Vit.A	BP	IBD	51 IBD (ns); 67 AD (90% tubulare); 136 CRC; 79 HC	↑ in IBD vs. CRC	Diff. IBD/CRC	[89]
PC	T-H	CD, UC	22 UC (15/ns), paired biopsies inf. & non-infl. colon (*n* = 6 aUC); 11 CD (6/ns); 14 specific colitis (infl.); 10 GIS	↑ in aUC and iUC vs. GIS; ↑ in aCD and iCD vs. GIS; ↑ in spec. colitis vs. GIS; ↑ in aUC/aCD vs. iUC/iCD; ↑ in infl. and non-infl. aUC vs. iUC and GIS; ↑ in infl. aUC vs. non-infl.; If stratified by severity into: CON/iIBD/miIBD/moIBD/sIBD, r = 0.81	Diff. IBD/GIS	[19]
nTyr	T-H	CD, UC	22 UC (15/ns), paired biopsies inf. & non-infl. colon (*n* = 6 aUC); 11 CD (6/ns); 14 specific colitis (infl.); 10 GIS	↑ in aUC and iUC vs. GIS; ↑ in aCD and iCD vs. GIS; ↑ in spec. colitis vs. GIS; ↑ in aUC/aCD vs. iUC/iCD; ↑ in infl. and non-infl. aUC vs. iUC and GIS; ↑ in infl. aUC vs. non-infl.; If stratified by severity into: CON/iIBD/miIBD/moIBD/sIBD, r = 0.84	Diff. IBD/GIS	[19]
8-OHdG	BL	IBD	51 IBD (ns); 67 AD (90% tubulare); 136 CRC; 79 HC	↑ in IBD vs. AD and CRC and HC	Diff. IBD/N	[89]

^1^ number of patients (number of patients with active disease/scoring system ≥ cut-off for active disease); ^2^ data presented as ↑ increased or ↓ decreased levels between indicated groups and as correlation coefficients preceded by the variable; ^3^ data showing fold change in expression and analyzed separately for three bowel fragments: AsC = ascending colon, DeS = descending colon, TI = terminal ileum; ns, not specified; aa, all active; *n*/a, no association; inflam., inflamed; IoFRP, intensity of free radical processes determined as sum of spontaneous chemiluminescence; RoPRG, rate of peroxide radical generation; NO, nitric oxide; LOX, lipoxygenase; NOS2, inducible NO synthase; TAS, total antioxidant status; S_IND_2, the sum of light energy over 2 min depends on activity of the antioxidant and antiradical defense system; FRAP, assay which measures the reduction of Fe^3+^ (ferric ion) to Fe^2+^ (ferrous ion) in the presence of antioxidants; t-bil, total bilirubin;alb, albumin; CP_P_, ceruloplasmin determined as protein; StP, susceptibility to undergo peroxidation; 8-OHdG, 8-oxo-2’-deoxyguanosine; nTyr, 3-nitrotyrosine; 8-iso-PGF2a, 8-iso-prostaglandin F2 alpha; LOOH, lipid hydroperoxides; Vit, vitamin; GSH, reduced glutathione; CD, Crohn’s disease; UC, ulcerative colitis; IBD, inflammatory bowel disease; iIBD, inactive IBD; miIBD, mild IBD; moIBD, moderate IBD; sIBD, severe IBD; aCD, active CD; iCD, inactive CD; aUC, active UC; iUC, inactive UC; GIS, non-IBD patients with gastrointestinal symptoms; IBS, irritable bowel syndrome; CRC, colorectal cancer; AD, adenoma; N, neoplasms; HC, healthy control; Diff., differential marker; FFPE, formalin-fixed paraffin-embedded; T-E, tissue-based marker analyzed as mRNA expression; T-M, tissue metaproteomics—aspirates collected from mucosal-luminal interface for the analysis of microbial and human proteins; BS, blood-based marker determined in serum; BP, blood-based marker determined in plasma; T-H, tissue-based marker determined in homogenates; PBMC, peripheral blood mononuclear cells; Diff_IBD/GIS_, differential marker for IBD and GIS; Diff_CD/IBS_, differential marker for CD and IBS; Diff_CD/UC_, differential marker for CD and UC; CDAI, Crohn’s disease activity index; PCDAI, pediatric Crohn’s disease activity index; PUCAI, pediatric ulcerative colitis index; IS, interventional study.

**Table 3 diagnostics-10-00601-t003:** Oxidative stress indices as potential biomarkers of progression in inflammatory bowel diseases.

Analyte	Type	Disease	Evaluated Population ^1^	Findings ^2^	Characteristics	Ref.
NOS2	T-I	UC	141 FFPE rectal UC mucosa slides: 18 N (9 D and 9 UCAC)	Insignificantly higher IHC-score in UC+N vs. UC alone	Progression marker: cut-of = 50 IHC-score: sens. 67%, spec. 49%	[113]
CAT	BE	IBD	93 UC (42/RI, ns) 81 CD (53/CDAI ≥ 150); 105 HC	↓ in aCD and aUC vs. HC; ↓ in aCD vs. iCD; CDAI −0.30; CRP ^3^ −0.34; ESR ^3^ −0.32; MES ^4^ −0.22; ESR ^4^ −0.22	Active IBD marker: AUC = 0.63 ^5^; cut-off ≤128 U/mL: sens. 86%, spec. 39%	[31]
GPx	BE	IBD	93 UC (42/RI, ns) 81 CD (53/CDAI ≥ 150); 105 HC	↓ in aCD and iCD vs. HC; ↓ in aUC and iUC vs. HC; ↓ in aCD vs. iCD; ↓ in aUC vs. iUC; CDAI −0.50; CRP ^3^ −0.36; IL-6 ^3^: −0.42; RI −0.36; ESR ^4^ −0.37	Active IBD marker: AUC = 0.73 ^5^; Cut-off ≤5.88 U/mL: sens. 58%, spec. 80%	[31]
PON1_P_	BP	CD	47 CD (28/CDAI≥150); 23 HC	↓ in aCD vs. iCD; CRP: −0.61	Active CD marker: AUC = 0.87; cut-off 305.31 U/L, sens. 86%, spec. 84%	[47]
F-SH	BS	IBD	47 CD (37/CDAI ≥ 150) 71 UC (30/RI ≥ 6); 57 HC	↓aCD and iCD vs. HC; ↓aUC and iUC vs. HC; CDAI −0.52; RI −0.40; MES ^4^ −0.27; CRP ^3^ −0.56; ESR ^3^ −0.51; IL-6 ^3^ −0.45	MH (MES = 0–1) marker; AUC = 0.69; >409 µM, sens. 74% and spec. 60%	[73]
Trf	BS	IBD	63 CD (36/CDAI ≥ 150) 74 UC (28/RI ≥ 6); 97 HC	↓aCD and aUC vs. iCD and iUC and HC; CDAI −0.49; CRP ^3^ −0.52; IL-6 ^3^ −0.47; RI −0.52; MES ^4^ −0.37; CRP ^4^ −0.29; ESR ^4^ −0.31; IL-6 ^4^ −0.49	Active UC marker: AUC = 0.76 (0.65–0.85); cut-off ≤279 mg/dL, sens. 86%, spec. 65%	[56]
TAS (ABTS)	BS	IBD	47 CD (37/CDAI ≥ 150) 71 UC (30/RI ≥ 6); 57 HC	↓aCD and iCD vs. HC; ↓aUC and iUC vs. HC; CDAI −0.34; MES ^4^ −0.58; CRP ^3^ −0.56; CRP ^4^ −0.60	MH (MES = 0–1) marker: AUC = 0.87; cut-off>1.4 mM, sens. 61%, spec. 100%	[73]
TAS (FRAP)	BS	CD	58 CD (36/CDAI ≥ 150); 26 HC	↓aCD vs. iCD and HC; CDAI −0.8; CRP −0.5	Active CD marker: AUC = 0.94 (0.89–0.99) ^7^	[37]
UA	BS	IBD	47 CD (37/CDAI ≥ 150) 71 UC (30/RI ≥ 6); 57 HC	↓aCD and aUC vs. iCD and iUC and HC; CDAI −0.35; MES ^4^ −0.41	MH (MES = 0–1) marker: AUC = 0.77; cut-off>2.81 mg/dL, sens. 76%, spec. 80%	[73]
GSH (DTNB)	BS	CD	58 CD (36/CDAI ≥ 150); 26 HC	↓aCD vs. iCD and HC; CDAI −0.8; CRP −0.5	Active CD marker: AUC = 0.86 (0.78–0.95) ^4^	[37]
MDA (TBARS)	BS	CD	58 CD (36/CDAI ≥ 150); 26 HC	↑aCD vs. iCD and HC; CDAI 0.8; CRP 0.6	Active CD marker: AUC = 0.95 (0.9–1) ^4^; cut-off >3.82 nmol/g, sens. 93%, spec. 80%	[37]
8-OHdG	T-H	UC	Multiple biopsies from: 50 UC patients, 6 UC+LGD, 4 UC+HGD/UCAC, 5 sporadic CRC, 11 GIS; BES	↑in UC and UC+LGD vs. GIS; ↑ in UC+LGD vs. UC; ↑ in SC vs. R;HA: r = 0.33; EA: r = 0.37	Progression marker: AUC = 0.81 (SE = 0.06); cut-off: 100 adducts/10^5dG, sens. = 70%, spec. = 78%, PPV = 47%, NPV = 94%	[114]
8-OHdG	T-I	UC	141 FFPE rectal UC mucosa slides: 18 N (9 dysplasia and 9 UCAC)	↑ IHC-score in UC+N vs. UC alone	Progression marker: cut-off = 196 IHC-score: sens. 89% and 55% spec.	[113]
8-NG	T-I	UC	141 FFPE rectal UC mucosa slides: 18 N (9 D and 9 UCAC)	↑ IHC-score in UC+N vs. UC alone	Progression marker: cut-off = 52 IHC-score: sens. 83% and 83% spec.	[113]

^1^, number of patients (number of patients with active disease/scoring system≥cut-off for active disease); ^2^, data presented as ↑ increased or ↓decreased levels between indicated groups and as correlation coefficients preceded by the variable; ^3^, correlation in CD patients; ^4^, correlation for UC patients; ^5^, performance similar to that of C-reactive protein (CRP); ^6^, performance worse than that of CRP; ^7^, performance better than that of CRP; ns, not specified; NOS2, inducible nitric oxide synthase; CAT, catalase; GPx, glutathione peroxidase; PON1, paraoxonase-1 (A in superscript indicates the arylesterase activity and B – paraoxonase activity); F-SH, free thiols; TrF, transferrin; TAS (ABTS), total antioxidant status determined using ABTS assay; TAS (FRAP), total antioxidant status determined using FRAP method; UA, uric acid; GSH (DTNB), glutathione determined using DNTB method; MDA (TBARS), malondialdehyde determined as thiobarbituric acid-reactive substances; 8-OHdG, 8-hydroxy-2’ –deoxyguanosine; 8-NG, 8-nitroguanine; HS, healthy controls; CD, Crohn’s disease; UC, ulcerative colitis; IBD, inflammatory bowel diseases; aCD, active CD; iCD, inactive CD; aUC, active UC; iUC, inactive UC; N, neoplasms; D, dysplasia; LGD, low grade dysplasia; HGD, high grade dysplasia; UCAC, UC-associated cancer, CRC, colorectal cancer; R, rectum; GIS, non-IBD patients with gastrointestinal symptoms; CDAI, Crohn’s disease activity index, MES, Mayo endoscopic subscore; RI-EAI, Rachmilewitz endoscopic activity index; RI, Rachmilewitz index; HA, histologic activity; EA, endoscopic activity; FFPE, formalin-fixed paraffin-embedded; ESR, erythrocyte sedimentation rate; MH, mucosal healing; BES, Baron endoscopic score; T-I, tissue-based marker determined with immunohistochemistry (IHC); BS, blood-based marker determined in serum; BE, blood –based marker determined in erythrocytes; BP, blood-based marker determined in plasma; T-H, tissue-based marker determined in homogenates; ΔB, difference between alveolar breath and environmental air; AUC, area under receiver operating characteristics (ROC) curve; OR, odds ratios; B, regression coefficient; SE, standard error; sens., sensitivity; spec., specificity; PPV, positive predictive value; NPV, negative predictive value.

**Table 4 diagnostics-10-00601-t004:** Oxidative stress indices as prognostic biomarkers in inflammatory bowel diseases.

Analyte	Type	Disease	Evaluated Population ^1^	Findings ^2^	Ref.
Zn	BS	IBD	773 CD; 223 UC; patients stratified into Zn-deficient (Zn<0.66 mcg/ml) and Zn-normal; analysis of IBD registry data	Zn-deficient patients have ↑risk of: CD: hospitalizations OR = 1.44 (1.02–2.04), surgeries OR = 2.05 (1.38–3.05), complications OR = 1.50 (1.04–2.15); UC: hospitalizations, 2.14 (1.07–4.29), surgeries OR = 1.64 (0.59–4.52), complications OR = 1.97 (0.94–4.11)	[81]
MDA (TBARS)	BP	CD	25 adult CD; 88 HC;21 pediatric CD; 11 HC	↑in complications (abscess, fistula or stenosis)	[35]
GSH	BE	CD	25 adult CD; 88 HC; 21 pediatric CD; 11 HC	↓ in complications (abscess, fistula or stenosis)	[35]
t-bil	BS	IBD	242 CD; 211 UC; 255 HC	↓ in UC-E3 (extension); ↓ in colonic CD (L2); ↓ in penetrating CD(B3)	[75]

^1^, number of patients; ^2^, data presented as ↑ increased or ↓decreased levels between indicated groups and as correlation coefficients preceded by the variable; t-bil, total bilirubin; GSH (DTNB), glutathione determined using DNTB method; MDA (TBARS), malondialdehyde determined as thiobarbituric acid-reactive substances; Zn, zinc; HS, healthy controls; CD, Crohn’s disease; UC, ulcerative colitis; IBD, inflammatory bowel diseases; CDAI, Crohn’s disease activity index; MDAI, Mayo disease activity index; PCDAI, pediatric Crohn’s disease activity index; BS, blood-based marker determined in serum; BE, blood –based marker determined in erythrocytes; BP, blood-based marker determined in plasma; OR, odds ratios.

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
