# Peer review of "Oxidative Stress Markers in Inflammatory Bowel Diseases: Systematic Review"

_diagnostics, 2020, doi:10.3390/diagnostics10080601_

Round 1
Reviewer 1 Report
This systematic review aims to provide a comprehensive overview of oxidative stress markers as potential diagnostic, differential, progression, and prognostic markers in IBD.
Major points:
- Tables in their current form, although informative, are not comprehensive and hard to follow. These could be presented as supplementary material. I would suggest to include only four tables presenting only the most relevant clinical information of oxidative stress markers as potential diagnostic (Table 1), differential (Table 2), progression (Table 3), and prognostic markers (Table 4) in IBD. Three separate sections on each table should be also provided, one for CD, one for UC and one for IBD (if a marker is not specific for either CD or UC).
- The text should also be divided in subsections regarding the role of oxidative stress markers as potential diagnostic, differential, progression, and prognostic markers in IBD distinguishing to CD, UC and IBD (if a marker is not specific for either CD or UC) and not based on the different categories of the oxidative stress markers.
Author Response
Response to the Reviewer
This systematic review aims to provide a comprehensive overview of oxidative stress markers as potential diagnostic, differential, progression, and prognostic markers in IBD.
Major points:
Tables in their current form, although informative, are not comprehensive and hard to follow. These could be presented as supplementary material. I would suggest to include only four tables presenting only the most relevant clinical information of oxidative stress markers as potential diagnostic (Table 1), differential (Table 2), progression (Table 3), and prognostic markers (Table 4) in IBD. Three separate sections on each table should be also provided, one for CD, one for UC and one for IBD (if a marker is not specific for either CD or UC).
We would like to thank Reviewer for the detailed evaluation of our paper and valuable remarks. Tables in the current form are currently presented as supplementary materials (Tables S1-S8). As suggested, we included four tables reviewing oxidative stress markers as potential diagnostic (Table 1; line 500), differential (Table 2; line 629), progression (Table 3; line 924) and prognostic indices (Table 4; line 977) in the main body of the manuscript.
The text should also be divided in subsections regarding the role of oxidative stress markers as potential diagnostic, differential, progression, and prognostic markers in IBD distinguishing to CD, UC and IBD (if a marker is not specific for either CD or UC) and not based on the different categories of the oxidative stress markers.
We would like to thank Reviewer for this significant remark on our paper and we totally agree with the suggested changes. We reorganised our paper and divided the manuscript into recommended clinical subsections (diagnostic markers – line 153; differential markers – line 522; progression markers - 651; prognostic markers – line 949) instead of subsections based on the different types of the oxidative stress markers.
Reviewer 2 Report
Interesting review by Małgorzata Krzystek-Korpacka, Radoslaw Kempinski, Mariusz Bromke and Katarzyna Neubauerthat encompasses the knowledge considering oxidative stress markers that were examined towards their possibility to serve as biomarkers in inflammatory bowel diseases (IBD). The authors focused on the connection between IBD and oxidative stress. Moreover the review combines the knowledge of traditional medicine with experimental data.
Minor concerns:
Provide the enzyme commission number for all enzymes.
Provide incidence and prevalence of IBD in developing and developed countries and continents (graphs should be fine)
Line 455 - low molecular weight antioxidants?
Serum protein thiol levels in blood are a direct measure of the in vivo redox status in humans, because thiols react readily with oxygen-containing free radicals to form disulfides and overproduction of ROS in IBD can be quantified by measuring these specific components of systemic redox status. Please add some information about it.
In Conclusions section I'm missing the division of the stress markers for the type of tissue being examined (urine, blood, etc.) and invasive/non-invasive methods. Moreover, the last sentence is depressing, please add some future perspective.
Author Response
Response to the Reviewer 2
Interesting review by Małgorzata Krzystek-Korpacka, Radoslaw Kempinski, Mariusz Bromke and Katarzyna Neubauer that encompasses the knowledge considering oxidative stress markers that were examined towards their possibility to serve as biomarkers in inflammatory bowel diseases (IBD). The authors focused on the connection between IBD and oxidative stress. Moreover the review combines the knowledge of traditional medicine with experimental data.
We would like to thank the Reviewer for the kind words on our work.
Minor concerns:
Provide the enzyme commission number for all enzymes.
We would like to thank for this valuable remark. The enzyme commission numbers have been provided. All marked in green.
Provide incidence and prevalence of IBD in developing and developed countries and continents (graphs should be fine)
We would like to thank for this remark. We supplemented our paper with the suggested epidemiological data in lines 43-49 (marked in green)
Line 455 - low molecular weight antioxidants?
Subsections entitled low molecular weight antioxidants were removed from the manuscript.
Serum protein thiol levels in blood are a direct measure of the in vivo redox status in humans, because thiols react readily with oxygen-containing free radicals to form disulfides and overproduction of ROS in IBD can be quantified by measuring these specific components of systemic redox status. Please add some information about it.
We would like to thank the Reviewer for this remark. We added recommended information in lines 237-267 (marked in green).
In Conclusions section I'm missing the division of the stress markers for the type of tissue being examined (urine, blood, etc.) and invasive/non-invasive methods. Moreover, the last sentence is depressing, please add some future perspective.
We would like to thank the Reviewer for this precious remark. We supplemented this section with the data regarding studied tissues in lines 1061-1063, we removed the last sentence, as well as we added some future perspective, as suggested in lines 1080-1082 (marked in green).
Round 2
Reviewer 1 Report
The authors have adequately responded to the reviewers' comments.